# NOVA2-mediated RNA regulation is required for axonal pathfinding during development

Yuhki Saito[1], Soledad Miranda-Rottmann[1†‡], Matteo Ruggiu[1†§], Christopher Y Park[2], John J Fak[1], Ru Zhong[1], Jeremy S Duncan[3¶], Brian A Fabella[4], Harald J Junge[5], Zhe Chen[5], Roberto Araya[6‡], Bernd Fritzsch[3], A J Hudspeth[4], Robert B Darnell[1,2*]

[1]Laboratory of Molecular Neuro-Oncology, Howard Hughes Medical Institute, The Rockefeller University, New York, United States; [2]New York Genome Center, New York, United States; [3]Department of Biology, College of Liberal Arts and Sciences, University of Iowa, Iowa City, United States; [4]Laboratory of Sensory Neuroscience, Howard Hughes Medical Institute, The Rockefeller University, New York, United States; [5]Department of Molecular, Cellular, and Developmental Biology, University of Colorado, Boulder, Boulder, United States; [6]Department of Neurosciences, Faculty of Medicine, University of Montreal, Montreal, Canada

*For correspondence: darnelr@ rockefeller.edu

†These authors contributed equally to this work

Present address: ‡Department of Neurosciences, Faculty of Medicine, University of Montreal, Montreal, Canada; §Department of Biological Sciences, St. John's University, Utopia Parkway, United States; ¶Division of Otolaryngology, University of Utah, Salt Lake city, United States

Competing interests: The authors declare that no competing interests exist.

**Abstract** The neuron specific RNA-binding proteins NOVA1 and NOVA2 are highly homologous alternative splicing regulators. NOVA proteins regulate at least 700 alternative splicing events in vivo, yet relatively little is known about the biologic consequences of NOVA action and in particular about functional differences between NOVA1 and NOVA2. Transcriptome-wide searches for isoform-specific functions, using NOVA1 and NOVA2 specific HITS-CLIP and RNA-seq data from mouse cortex lacking either NOVA isoform, reveals that NOVA2 uniquely regulates alternative splicing events of a series of axon guidance related genes during cortical development. Corresponding axonal pathfinding defects were specific to NOVA2 deficiency: *Nova2-/-* but not *Nova1-/-* mice had agenesis of the corpus callosum, and axonal outgrowth defects specific to ventral motoneuron axons and efferent innervation of the cochlea. Thus we have discovered that NOVA2 uniquely regulates alternative splicing of a coordinate set of transcripts encoding key components in cortical, brainstem and spinal axon guidance/outgrowth pathways during neural differentiation, with severe functional consequences in vivo.

## Introduction

During central nervous system (CNS) development, a neuron extends its axon through a complex yet precise path to reach its final destination by sensing extracellular signals called guidance cues. These cues are sensed by the growth cone, a motile structure at the extending axon edge, and they control growth cone motility through directed cytoskeletal remodeling. Netrins, slits, semaphorins, and ephrins are the major classic guidance cues and elicit attractive or repulsive responses in growth cones via specific receptors (*Brose et al., 1999*; *Cheng et al., 1995*; *Drescher et al., 1995*; *Fan and Raper, 1995*; *Kapfhammer and Raper, 1987*; *Kennedy et al., 1994*; *Kidd et al., 1999*; *Serafini et al., 1994*). An important aspect of axon guidance is the spatial and temporal control of response to the guidance cues. For example, the spinal cord commissural axon reaching the midline senses netrin-1, secreted from the floorplate as a chemoattractive cue; however, once it has crossed the floorplate, this cue becomes repulsive (*Kennedy et al., 1994*; *Kidd et al., 1998*; *Tessier-*

**eLife digest** The first step of producing a protein involves the DNA of a gene being copied to form a molecule of RNA. This RNA molecule can often be processed to create several different "messenger" RNAs (mRNAs), each of which are used to produce a different protein by a process known as alternative splicing. A class of proteins that bind to RNA molecules controls alternative splicing. These "splicing factors" ensure that the right protein variant is produced at the right time and in the right place to carry out the appropriate activity.

Many genes that play important roles in the nervous system have been reported to undergo alternative splicing to generate different protein variants. However, it is unclear whether alternative splicing is important for controlling how the nervous system develops, during which time the neurons connect to the cells that they will communicate with. Forming these connections involves part of the neuron, called the axon, growing along a precise path through the nervous system to reach its destination.

Two RNA-binding proteins called NOVA1 and NOVA2 are produced exclusively in the central nervous system, where they regulate a number of actions including alternative splicing. So far, no differences in the roles of NOVA1 and NOVA2 have been identified, and relatively little is known about their actions in the brain.

Saito et al. have addressed these missing puzzle pieces by combining RNA analysis methods with an analysis of the structure of the nervous system of mice that lack either NOVA1 or NOVA2. This approach identified where NOVA1 and NOVA2 bind on mRNAs, and showed that the mRNAs are processed in different ways in the developing mouse brain depending on which form of the NOVA protein is bound to it.

Further analysis of the data revealed that NOVA2, and not NOVA1, regulates splicing in a series of RNA molecules that help to guide axons to the correct locations in the developing mouse brain. A related study by Leggere et al. also reported on the role that NOVA proteins play in the alternative splicing of one of these genes, called *Dcc*.

Saito et al. also found defects in the nervous systems of the mice that lacked NOVA2 that only occurred in these mice and resulted from certain axons being unable to follow the correct path to their target cells. These led to major defects, such as agenesis of the corpus callosum (a complete lack of connection between the right and left sides of the brain). Further defects affected how specific subsets of motor neurons connect to muscles and how cochlear neurons in the brainstem connect to the inner ear. The next steps are to explore how the processing of RNA molecules by NOVA2 causes these defects, and to assess whether these actions relate to developmental brain disorders in humans.

*Lavigne et al., 1988*; *Zou et al., 2000*). Furthermore, the spatiotemporally restricted expression of Robo3 alternative splicing isoforms in spinal cord commissure axons are essential for the switching of the growth cone response to the axon guidance cues (*Chen et al., 2008*), indicating that spatio-temporally regulated protein isoform expression and diversity is crucial to establish proper neuronal networks.

Alternative splicing and alternative polyadenylation can produce multiple messenger RNAs (mRNAs) possessing distinct coding and regulatory sequences from a single gene. The regulated processes that generate such mRNA diversity are orchestrated by RNA-binding proteins (RBPs). In the nervous system, alternative splicing has many important roles, including controlling the spatial and temporal expression of protein isoforms that are necessary for neurodevelopment and the modification of synaptic plasticity (*Li et al., 2007*; *Licatalosi et al., 2008*; *Ule and Darnell, 2006*). Significantly, human genetic studies have indicated that RNA misregulation resulting from defects in RBP expression and function are linked to numerous diseases, including Fragile X syndrome, spinal muscular atrophy, spinocerebellar ataxias, motoneuron disease and others (*Cooper et al., 2009*; *Darnell, 2010*; *Lukong et al., 2008*).

NOVA1 and NOVA2, RBPs initially identified as targets in autoimmune motor neuron disease (*Buckanovich et al., 1993*; *Darnell and Posner, 2003*), are RNA-binding splicing regulators

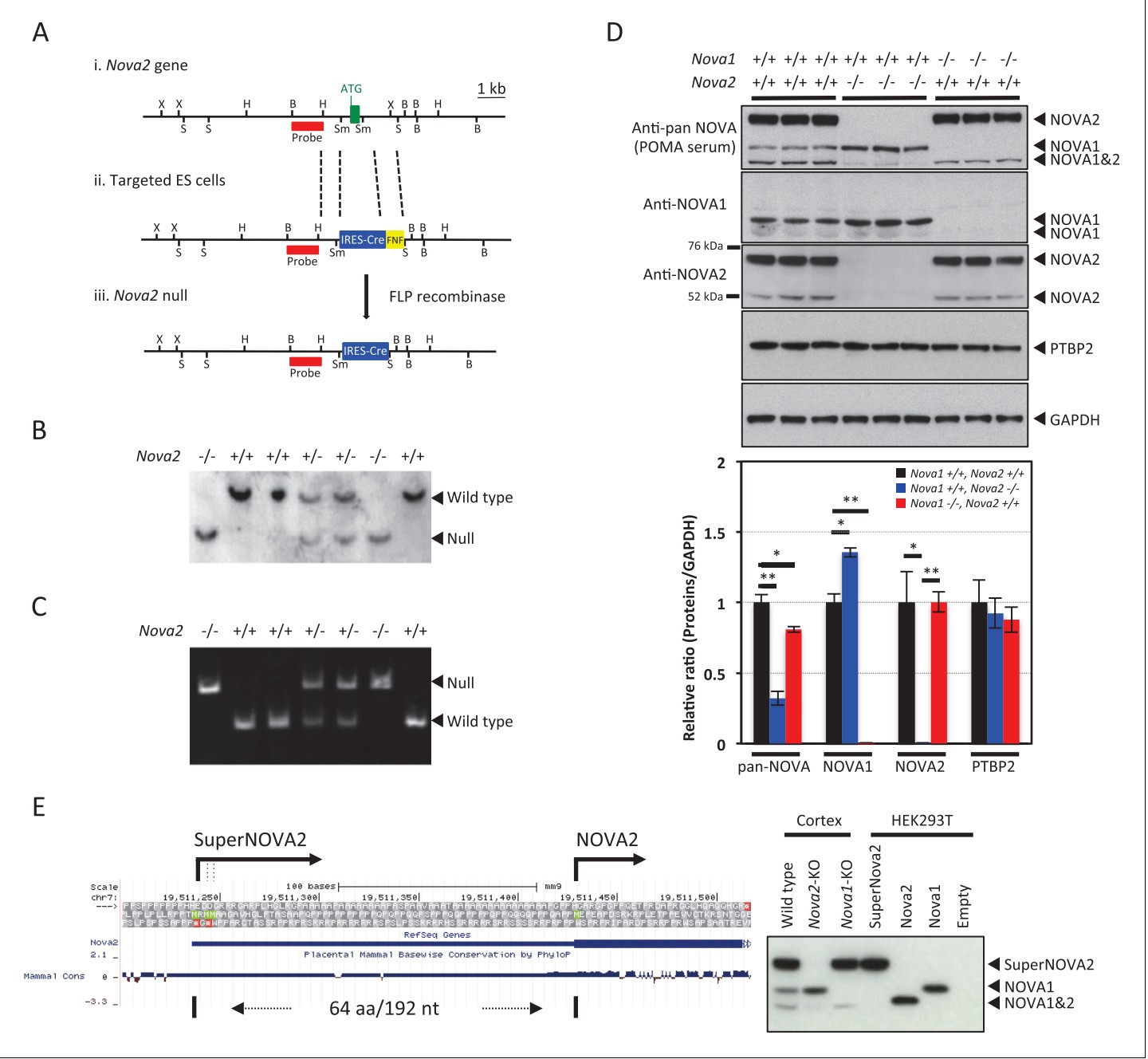

**Figure 1.** Generation of *Nova2* null mice and characterization of SuperNOVA2. (**Ai**) The wild-type *Nova2* locus illustrated contains the first exon (green box, with initiator ATG indicated). (**Aii**) A targeting construct was generated harboring a genomic fragment (left: 2.2 kb) flanking the initiator methionine, an *IRES-Cre FRT-NEO-FRT* (*FNF*) insertion, and an intronic genomic fragment flanking the first coding exon (right: 6 kb). (**Aiii**) The *Nova2* null locus following FLP-mediated excision of *FNF* cassette. Restriction enzyme sites were indicated for BamHI (**B**), HindIII (**H**), SacI (**S**), SmaI (**Sm**) and XbaI (**X**). The probes position used for Southern blot was indicated in *red*. (**B**) Genotypic analysis of *Nova2* null mice. Southern blot analysis was performed on tail DNA digested with BamHI, using the probe described in (**A**). (**C**) Genotyping PCR analysis of *Nova2* null mice. (**D**) Western blot analysis of NOVA1 and NOVA2 proteins. Extracts of mouse cortex (10 μg/lane) were made from age-matched P0 wild-type, *Nova2-/-*, and *Nova1-/-* mice, loaded on SDS-PAGE gels, and blotted with anti-pan NOVA (POMA antisera), anti-NOVA1 specific, anti-NOVA2 specific, anti-PTBP2, and anti-GAPDH antibodies. Quantification and comparison of NOVA1 and NOVA2 proteins expression amounts in the cortex of wild-type, *Nova2-/-*, and *Nova1-/-* mice. Data are presented as mean ± SD. *p<0.05, **p<0.01 (n = 3, Tukey's multiple comparison test). (**E**) Characterization of superNOVA2. *Left Diagram* showing a putative superNOVA2 initiator methionine (*green*) positioned 192 nt upstream of the known *Nova2* initiator methionine. *Right panel* showing the NOVA proteins mobility on electrophoresis.

*Figure 1 continued on next page*

*Figure 1 continued*

The following figure supplements are available for figure 1:

**Figure supplement 1.** Growth retardation of *Nova2-/-* mice.

**Figure supplement 2.** Anti-NOVA2 and anti-NOVA1 antibodies specificity.

specifically expressed in neurons in the CNS (*Buckanovich et al., 1996*; *Yang et al., 1998*). NOVA1 and NOVA2 harbor three KH-type RNA binding domains, and in vitro RNA selection (*Buckanovich and Darnell, 1997*; *Jensen et al., 2000b*; *Yang et al., 1998*) and X-ray crystallography (*Lewis et al., 2000*; *Teplova et al., 2011*) demonstrate that NOVA1 and NOVA2 bind directly to RNA sequences harboring YCAY motifs. The development of RNA:RBP crosslinking and immunopre-cipitation (CLIP) (*Ule et al., 2003*; *2005a*) followed by high-throughput sequencing (HITS-CLIP) (*Licatalosi et al., 2008*) methods has enabled identification of genome-wide RBP:RNA interaction maps in vivo. The bioinformatic integration of HITS-CLIP binding data and functional outcome revealed by exon junction microarray and RNA-seq data sets led to the conclusion that NOVA regu-lates alternative splicing following the rule that the position of NOVA binding to pre-mRNA predicts its action to enhance or inhibit alternate exon inclusion (*Licatalosi et al., 2008*; *Ule et al., 2005b*; *2006*; *Wu et al., 2013*, *Zhang et al., 2010*). These analyses revealed ~700 NOVA1/NOVA2 target alternate exons and allowed to predict the biological process regulated by NOVA1/NOVA2. However, these studies have found no biochemical actions that were unique to NOVA1 versus NOVA2 paralogues, reflecting their nearly identical KH-type RNA binding domains.

In the present work, we find that NOVA2 uniquely regulates a series of alternative splicing events of axon guidance related genes, including deleted in colorectal carcinoma (*Dcc*), Roundabout, Axon Guidance Receptor, Homolog 2 (*Robo2*), Slit homolog 2 (*Slit2*), and EPH Receptor A5 (*Epha5*). These findings derived from combining transcriptome-wide NOVA1 and NOVA2 specific HITS-CLIP with RNA-seq analysis in the cortex of mice lacking either *Nova1* or *Nova2*. *Nova2* deficiency results in some common phenotypes, such as severe growth retardation, as previously reported in *Nova1-/-* mice (*Jensen et al., 2000a*), as well as unique phenotypes, including agenesis of the corpus callosum (ACC) in *Nova2-/-* mice. Unexpectedly, given that motoneurons express both NOVA1 and NOVA2, a subset of motoneuron axons directed to the ventral diaphragm show outgrowth defects in *Nova2-/-* but not *Nova1-/-* mice. In addition, the efferent innervation to the cochlea which is derived during development from a ventral subpopulation of facial motoneurons, is normal in *Nova1-/-* mice, reduced in *Nova2-/-* mice, and completely stalled in the absence of both *Nova1* and *Nova2*, sug-gesting that alternative splicing regulator NOVA2 plays critical roles in axon pathfinding context in mammals and that NOVA1 may have a cooperative role in this process. Based on these observations from genome-wide and histological analyses of *Nova2-/-* mice, we conclude that NOVA2-mediated RNA regulation is essential for CNS development by regulating neural networks wiring.

## Results

### Generation of *Nova2* null mice and growth retardation in *Nova2* null mice

We utilized targeted mutagenesis to disrupt *Nova2* genetic function in mice. A targeting cassette was designed to replace the first exon of *Nova2* gene (*Figure 1A*). The genotype of the resulting progenies was confirmed both by Southern blot (*Figure 1B*) and PCR (*Figure 1C*). The progeny dis-played the expected Mendelian ratio of mice for the heterozygous and homozygous *Nova2* muta-tion. To confirm that homozygous *Nova2* mutant mice did not express NOVA2 protein, Western blot analysis was carried out on protein extracts from P0 cortex (*Figure 1D*) of wild-type, *Nova2-/-*, and *Nova1-/-* mice, using anti-NOVA1, anti-NOVA2, and antisera from a patient with paraneoplastic opsoclonus-myoclonus ataxia (POMA), which recognizes all NOVA protein species (*Yang et al., 1998*). Expression of NOVA2 protein isoforms was absent in *Nova2-/-*mice (*Figure 1D*).

Interestingly, more than one protein isoform was absent in protein extracts from *Nova2* null mice: a single 50–52 kDa NOVA2 protein species corresponding to the predicted size of the previously

described *Nova2* ORF, and a series of previously described (*Ule et al., 2005a*; *Eom et al., 2013*) but uncharacterized protein isoforms of ~70 kDa recognized by anti-NOVA2 antibody and POMA anti-sera, which we named SuperNOVA2. We found three putative initiator ATG sequences upstream of the known *Nova2* start codon on the 5' UTR of the transcript. The mobility of SuperNOVA2 on electrophoresis was comparable with the protein product expressed from SuperNOVA2 plasmid containing full-length sequence of the 5' UTR (*Figure 1E*), confirming that SuperNOVA2 translation started from an upstream codon in the *Nova2* 5' UTR. Using anti-NOVA1 or SuperNOVA2/NOVA2 specific antibodies, we found that NOVA1 protein levels were significantly increased, by ~40% in *Nova2-/-* mice cortex (*Figure 1D*) but NOVA1 anatomic expression pattern was similar to wild-type mice sections (data not shown). Thus NOVA1 has partially overlapping distributions with SuperNOVA2/NOVA2 (hereinafter we defined SuperNOVA2/NOVA2 as NOVA2), is upregulated by unknown means in the absence of *Nova2*, and may have partially redundant functions with NOVA2 in the cortex.

*Nova2* null animals were born indistinguishable from littermates but failed to thrive, demonstrating progressive motor dysfunction and overt motor weakness, and they died an average of 14–18 days after birth (*Figure 1—figure supplement 1*). There was no apparent phenotype in *Nova2* heterozygotes, although we previously found that when assessed by electroencephalography, 6 month old *Nova2+/-* (and *Nova1+/-*) mice had frequent synchronous cortical interictal discharges consistent with epilepsy (*Eom et al., 2013*). In conclusion, we demonstrate that the *Nova2* gene encodes for multiple protein isoforms, including an identified ~70 kDa SuperNOVA2 isoform, and that *Nova2* is necessary for post-natal survival.

## Genome-wide NOVA1- and NOVA2-RNA interaction maps

The NOVA1/2 splicing-regulatory network and its target alternative splicing events have been previously defined by utilizing NOVA2 and pan-NOVA HITS-CLIP data, splicing sensitive microarray data, NOVA1/2-binding motifs, and integrative modeling (*Licatalosi et al., 2008*; *Ule et al., 2005b*; *2006*; *Zhang et al., 2010*), yet these data sets were integrated data sets from multiple developmental stages, CNS regions, and *Nova1/Nova2* genotypes. Relatively little is known about the biologic consequences of NOVA1 and NOVA2 action and about the functional differences between NOVA1 and NOVA2 in a specified CNS region and at a particular developmental stage.

To identify NOVA1 and NOVA2 direct RNA targets in E18.5 cortex, a transcriptome-wide library of NOVA1-RNA and NOVA2-RNA interactions were generated by HITS-CLIP from E18.5 wild-type mice cortex and sequenced. NOVA1 and NOVA2 CLIP reads were filtered and aligned to the mouse genome (mm9). A total of 2,318,553 unique NOVA2 CLIP tags were obtained from three biological replicates (483,446, 836,678, and 998,429 unique tags, respectively), with a total of 139,007 clusters from at least two or three biologic replicates (defined as biological complexity of two (BC2)). A total of 910,342 unique NOVA1 CLIP tags were obtained from three biological replicates (319,904, 292,709, and 297,729 unique tags, respectively), with a total of 54,546 clusters (BC2). To identify sequence features most commonly associated with either NOVA1-RNA or NOVA2-RNA interactions, 4 nucleotide (nt) long motifs were counted in BC2 NOVA1 and NOVA2 clusters and in a set of shuffled control sequences as a control. Each of the top five ranking tetramers in NOVA1 and NOVA2 clusters were the same (UCAU, CAUC, CAUU, UCAC, and UUCA) (*Figure 2A*) and coincided with the previously identified YCAY NOVA binding motif. Enrichment of the YCAY motif around NOVA1 and NOVA2 clusters was comparable between NOVA1 and NOVA2 and greatly enriched within NOVA1 and NOVA2 CLIP clusters (*Figure 2B*), suggesting that NOVA1 and NOVA2 recognized the same YCAY RNA sequence. NOVA1 and NOVA2 CLIP tag positions on known NOVA regulated alternatively spliced exons (*Zhang et al., 2010*) were comparable (*Figure 2C and D*). For example, alternative splicing of *Agrn* minigene containing exon 31–34 were regulated by NOVA1 and NOVA2 in the same splicing direction, in this case mediating the inclusion of the exon (*Figure 2—figure supplement 1*), indicating that NOVA1 and NOVA2 have similar RNA interaction profiles on NOVA1/2 target alternative splicing sites and are capable of regulating the same alternative splicing events.

Twenty-six percent or eighteen percent, respectively, of the NOVA2 or NOVA1 BC2 clusters consisted of a minimum of 10 tags (defined as peak height (PH); NOVA2; 36,591 of 139,007, NOVA1; 9988 of 54,546). 80.1% of NOVA1 clusters were shared with NOVA2 cluster positions, while 22.1% of NOVA2 clusters overlapped with NOVA1 (*Figure 2E*). Interestingly, the genomic distribution of NOVA2 clusters and NOVA1 showed different distributions within their target transcripts (*Figure 2F*

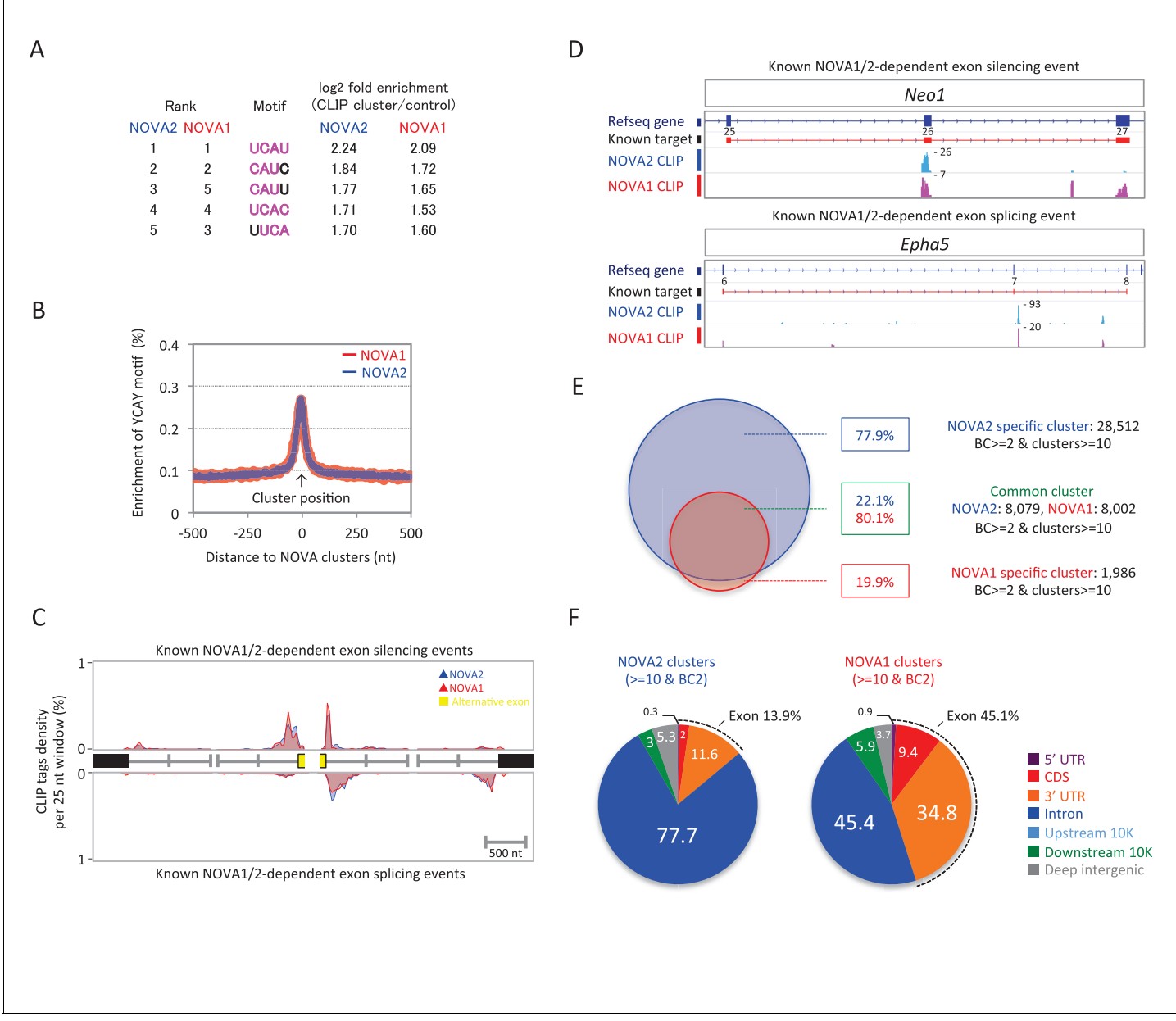

**Figure 2.** NOVA2 and NOVA1 HITS-CLIP. (**A**) Top 5 tetramers present in BC2 NOVA2 and NOVA1 sequences compared with the shuffled control sequences. The sequences corresponding to YCAY motif were high-lighted with *magenta*. (**B**) Enrichment of YCAY motif near NOVA2 (*blue*) and NOVA1 (*red*) clusters. (**C**) Distribution of BC2 NOVA2 (*blue*) and NOVA1 (*red*) CLIP tags (*Y*-axis: CLIP tags density in each 25 nt window) near known NOVA1/2 targeted alternative exons silencing (*top panel*) and splicing (*bottom panel*) events. (**D**) Examples of NOVA2 and NOVA1 CLIP cluster location near known NOVA1/2-target alternative splicing exon (*top panel*; silencing event, *Neo1* exon 25–27, *bottom panel*; splicing event, *Epha5* exon 6–8). (**E**) Summary of robust NOVA2 (*blue*) and NOVA1 (*red*) CLIP BC2 clusters consisted of a minimum of 10 tags. (**F**) Genomic distribution of BC2 NOVA2 (*left*) and NOVA1 (*right*) clusters consisted of a minimum of 10 tags.

The following figure supplements are available for figure 2:

**Figure supplement 1.** Alternative splicing regulation of *Agrn* minigene by NOVA proteins.

**Figure supplement 2.** Genomic distribution of BC2 NOVA2 specific (*left*), NOVA1 specific (*middle*), and NOVA2/NOVA1 common clusters consisted of a minimum of 10 tags.

**Figure supplement 3.** NOVA1 and NOVA2 expression in the mouse embryonic brain.

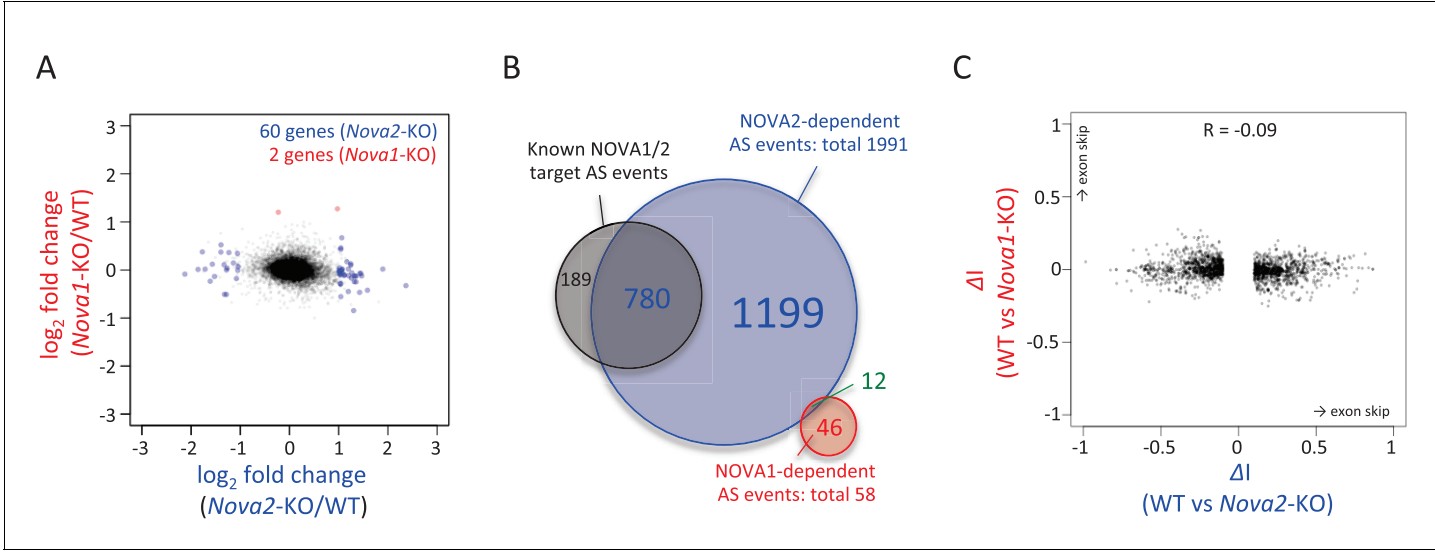

**Figure 3.** RNA-seq analysis in either *Nova2-/-* or *Nova1-/-* versus littermate wild-type control. (**A**) RNA abundance changes in *Nova2-/-* and *Nova1-/-* mice cortex and comparison of its correlation in *Nova2-/-* with in *Nova1-/-* mice. X-axis and Y-axis indicated log$_2$(*Nova2-KO*/wild-type fold change (FC)) and log$_2$(*Nova1-KO*/wild-type FC), respectively. Transcripts significantly changed in *Nova2-/-* and *Nova1-/-* mice cortex were shown in *blue* and *red*, respectively (FDR<0.05 and log$_2$|FC|>=1). (**B**) Summary of NOVA2- and NOVA1-dependent alternative splicing changes that showed |ΔI| >= 0.1 (FDR < 0.1) (*Ule et al., 2005b*) and that contained YCAY motif(s) within BC2 NOVA1 or NOVA2 HITS-CLIP clusters on alternative splicing exons and/or its upstream/downstream introns. 1991 NOVA2-target alternative splicing events on 540 genes (known [*Zhang et al., 2010*]; 780 events, novel; 1211 events), 58 NOVA1-target events on 20 genes (known; 0 events, novel 58 events). (**C**) Correlation of *Nova2-* and *Nova1-*deficient impact on alternative splicing events. X-axis and Y-axis indicated ΔI of wild-type (WT) vs *Nova2-/-* and vs *Nova1-/-*, respectively.

The following source data and figure supplements are available for figure 3:

**Source data 1.** Gene Ontology (GO) terms associated with NOVA2-target alternative splicing exons.

**Source data 2.** The transcriptome abundance of the neuronal differentiation markers in the E18.5 cortex of *Nova1-/-* and *Nova2-/-* mice.

**Figure supplement 1.** Summary of the number of alternative splicing events detected by RNA-seq analysis with/without NOVA2 or NOVA1 CLIP clusters and/or YCAY clusters.

**Figure supplement 2.** KEGG pathways over-represented among NOVA2-target genes.

**Figure supplement 3.** KEGG pathways over-represented among NOVA2-target genes.

**Figure supplement 4.** RNA-seq analysis in the midbrain and hindbrain of *Nova1-/-* versus littermate wild-type control.

and *Figure 2—figure supplement 2*). 77.7% of NOVA2 clusters were located in introns and 13.9% in exons (5' UTR, CDS, and 3' UTR) compared to 45.4% of NOVA1 clusters that were in introns and 45.1% in exons (*Figure 2F*). 84.9% of NOVA2 specific BC2 clusters (PH>=10) were enriched in introns, while 74.1% of NOVA1 specific BC2 clusters (PH>=10) were in exons (*Figure 2—figure supplement 2*), indicating that NOVA2 binds preferentially to introns than exons, suggesting that NOVA2 may play a greater nuclear role than NOVA1, and demonstrating that RNA-interaction profiles on a genome-wide scale are different between NOVA homologues.

The distribution of NOVA2 throughout the brain mirrored previous immunohistochemical and in situ hybridization data (*Yang et al., 1998*) showed that NOVA2 was expressed at high levels in cortex and hippocampus, and at lower levels in midbrain and spinal cord, where NOVA1 was expressed at high levels in a generally reciprocal fashion, with low levels in the cortex and relatively high levels in the midbrain and spinal cord (*Figure 2—figure supplement 3*). The NOVA2 expressed cell in the cortical plate of neocortex was ubiquitously distributed at comparable expression level, yet NOVA1 was expressed in the specified cell types. Taken together, the HITS-CLIP and immunohistochemical

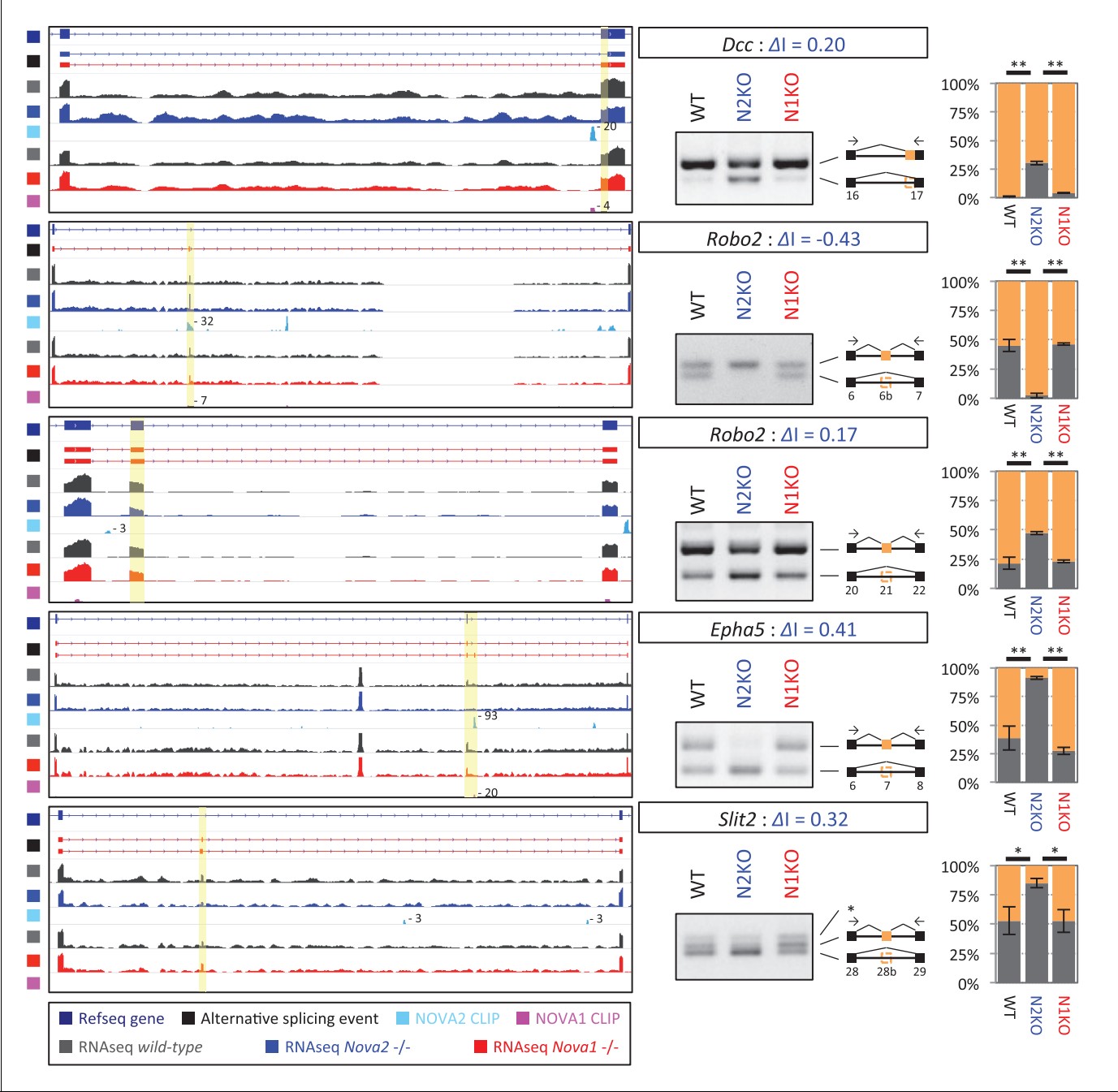

**Figure 4.** NOVA2 unique alternative splicing events of axon guidance related genes in E18.5 mice cortex. *Left Diagrams* showing RefSeq annotation genes, changed alternative splicing events, RNA-seq results of wild-type (*grey*) and *Nova2-/- (blue)*, NOVA2 CLIP clusters (*light blue*), RNA-seq results of wild-type (*grey*) and *Nova1-/- (red)*, and NOVA1 CLIP clusters (*pink*). *Right panels* and *graphs* showing RT-PCR results and quantification data in E18.5 wild-type, *Nova2-/-*, and *Nova1-/-* mice cortex, respectively. *p<0.05, **p<0.01 (n = 3, Tukey's multiple comparison test). Data are presented as mean ± SD.

The following figure supplement is available for figure 4:

**Figure supplement 1.** NOVA2 unique alternative splicing events of axon guidance related genes in E18.5 mice cortex.

data suggest that NOVA1 and NOVA2 perform unique biological functions in different brain areas and cell types, and that in those few cortical neurons expressing NOVA1, NOVA2 might be expected to have some redundant activity, while the reciprocal may not so often be the case.

## NOVA2-dependent alternative splicing changes of axon guidance related genes

To identify NOVA1 or NOVA2 target transcripts whose abundance or alternative splicing was changed in either *Nova1-/-* or *Nova2-/-* cortex, paired-end 125 nucleotide RNA-seq libraries were prepared from littermate E18.5 wild-type, *Nova1-/-, and Nova2-/-* mice cortex. A total of 225,054,059 and 213,596,515 unique reads were obtained from three biological replicates of wild-type and *Nova1-/-* mice, respectively, and a total of 190,282,241 and 180,200,564 unique reads were obtained from three biological replicates of wild-type and *Nova2-/-* mice, respectively. These experiments showed that the abundance of 60 transcripts were significantly changed in *Nova2-/-* mice compared to only 2 transcripts which showed steady-state changes in *Nova1-/-* mice compared to wild-type littermates (FDR<0.05 and |log$_2$FC|>=1) (*Figure 3A*), consistent with lower levels of NOVA1 expression in cortex. More changes were seen in analyzing splicing-dependent changes, and this pattern of differential NOVA2 and NOVA1 effects was also seen. We integrated NOVA1 and NOVA2 specific HITS-CLIP data, RNA-seq data, and NOVA binding YCAY motifs to define 1991 NOVA2 and 58 NOVA1-mediated alternative splicing events in the mouse cortex (*Figure 3B* and *Figure 3—figure supplement 1*). Interestingly, there was no correlation between *delta* I (ΔI) of wild-type versus *Nova1-/-* and wild-type versus *Nova2-/-* (R = 0.09) (*Figure 3C*). Taken together, this data shows that *Nova1* and *Nova2* have different RNA regulatory networks in developing cortical neurons; while the quantitative differences likely relate to the greater and more unique cellular expression of NOVA2 relative to NOVA1 in cortex, they also support the suggestion that the two NOVA homologues may regulate unique biological processes.

We explored the biologic pathways regulated by NOVA1 and NOVA2 with gene ontology (GO) analysis on the transcripts harboring NOVA-regulated alternative exons. Among the KEGG (Kyoto Encyclopedia of Genes and Genomes) pathway terms associated with NOVA2-regulated genes (FDR<0.05) were axon guidance signaling (FDR=0.0039) and adherens junctions signaling (FDR = 0.017) (*Figure 3—figure supplement 2* and *3*). In the GO terms associated with NOVA2-regulated genes (FDR<0.05), axonal projection related terms (cell morphogenesis, neuron projection morphogenesis, axonogenesis, cell morphogenesis involved in neuron differentiation, and cell projection morphogenesis) were also enriched (*Figure 3—source data 1*). NOVA2-dependent alternative splicing events of axon guidance related genes were validated by semi-quantitative RT-PCR with RNA prepared from E18.5 wild-type *Nova2-/-*, and *Nova1-/-* cortex (*Figure 4* and *Figure 4—figure supplement 1*). The alternative splicing events (*Dcc* exon 17, *Slit2* exon 28b, *Robo2* exons 6b and 21, *Epha5* exon 7, *Arhgef12* exon 4, *Ppp3cb* exon 10b, *Neo1* exon 26, and *Rock1* exon 27b) were significantly changed in *Nova2-/-* but not in *Nova1-/-* mice cortex, coinciding with RNA-seq results. No association was detected in the NOVA1-regulated alternative exons GO or KEGG pathways terms, likely due to the smaller number of NOVA1-regulated transcripts in cortex.

RNA-seq analysis in the E18.5 midbrain and hindbrain of wild-type and *Nova1-/-* mice, where NOVA1 is more abundantly expressed, identified that 119 of 1991 NOVA2-dependent alternative splicing events were changed in *Nova1-/-* mice (|ΔI|>=0.1, FDR<0.1) (*Figure 3—figure supplement 4A*). There was no correlation between ΔI of wild-type versus *Nova1-/-* (mid- and hindbrain) and wild-type versus *Nova2-/-* (cortex) (R=0.20) (*Figure 3—figure supplement 4B*) as well as identified between ΔI of wild-type versus *Nova1-/-* (cortex) and wild-type versus *Nova2-/-* (cortex). In the midbrain and hindbrain of *Nova1-/-* mice, only one alternative splicing event (*Robo2* exon 6b) was significantly changed in the NOVA2-regulated genes list of KEGG axon guidance pathway but it was smaller change when compared with ΔI of wild-type versus *Nova2-/-* (cortex) (*Figure 3—figure supplement 4C*). Taken together, these data suggest that NOVA2 regulates a distinct set of transcripts related, in particular, to axon guidance, in the E18.5 mouse cortex.

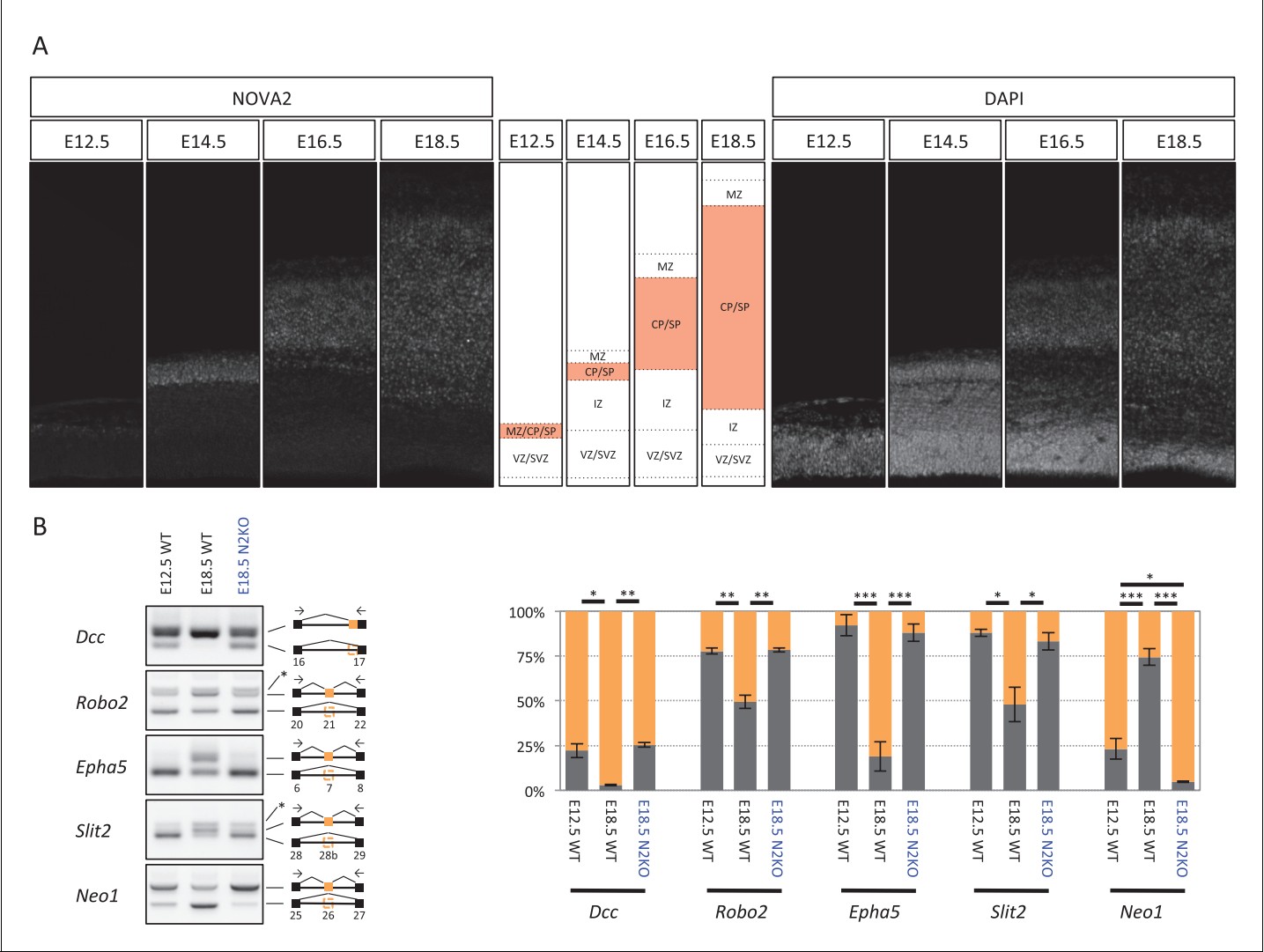

**Figure 5.** NOVA2 switches developmentally regulated exons usage of axon guidance related genes. (**A**) NOVA2 expression in developing neocortex (*left panels*). High NOVA2 expression regions were high-lighted with *red* in *middle panels*. MZ: marginal zone, CP: cortical plate, SP: subplate, IZ: intermediate zone, SVZ: subventricular zone, VZ: ventricular zone. (**B**) Analysis of NOVA2-regulated exons in *Dcc, Robo2, Epha5, Slit2*, and *Neo1* in E12.5 wild-type, E18.5 wild-type, and E18.5 *Nova2-/-* cortex (*left panels*). Quantification data of RT-PCR products were shown in *right graphs*. *$p<0.05$, **$p<0.01$, ***$p<0.001$ (n = 3, Tukey's multiple comparison test).

## NOVA2-mediated alternative splicing switch of axon guidance genes during mouse cortical development

To pursue the potential connection between NOVA2 and axonal guidance, we examined NOVA2 expression during cortical mouse brain development. We found that NOVA2 expression level detected by immunofluorescence was high in the cortical plate (CP) and subplate (SP) that consists of post-mitotic neurons at E18.5 (*Figure 2—figure supplement 3*). Similarly a high-expression of NOVA2 in CP and SP was also observed at E12.5, E14.5, and E16.5 (*Figure 5A*), indicating that cortical NOVA2 expression level was progressively up-regulated during neural differentiation from neural progenitor cells (NPC).

To assess whether NOVA2 regulated alternative splicing is also regulated in a developmentally regulated manner, we compared expression of alternatively spliced NOVA2 target exons in E12.5 wild-type, E18.5 wild-type, and E18.5 *Nova2-/-* cortex (*Figure 5B*). This revealed that NOVA2-target splicing events were developmentally regulated between E12.5 and E18.5, such that splicing in

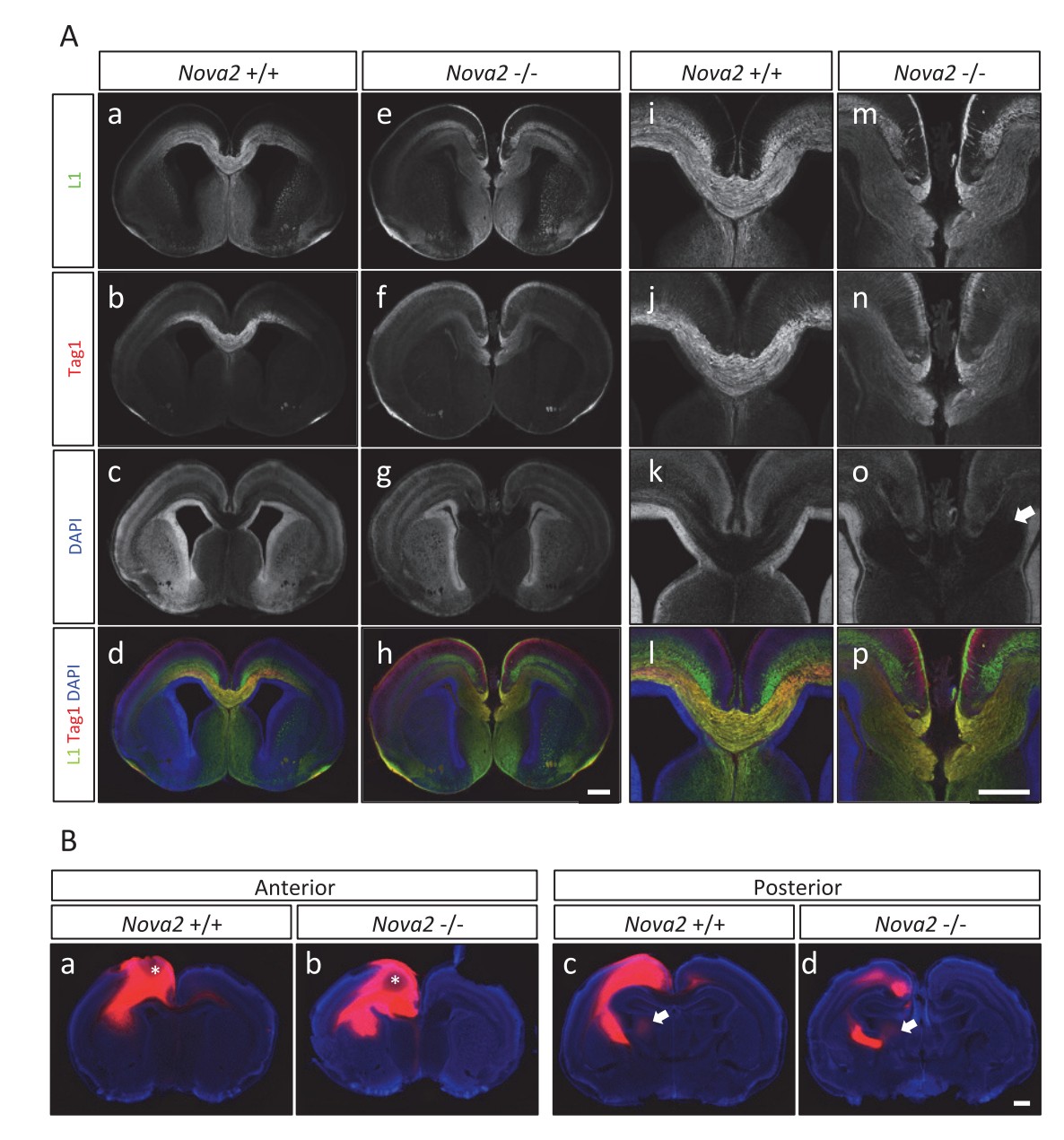

**Figure 6.** Agenesis of corpus callosum in Nova2-/- mice. (**A**) Immunohistochemistry of L1 (*green*; a,e,i,m), TAG1 (*red*; b,f,j,n) proteins, DAPI (*blue*; c,g,k, o) and merged views (d,h,l,p) on coronal sections in E18.5 wild-type (a–d, i–l) and *Nova2-/-* (e-h, m–p) littermates. (i–p) Higher magnified view of anterior commissure region of a–h. Arrows indicated Probst bundles. Scale bars; 500 μm. (**B**) Commissure axons pathfinding defect in *Nova2-/-* mice. Coronal sections of the anterior (a,b) and posterior (c,d) telencephalon in P0 wild-type (a,c) and *Nova2-/-* (b,d) mice showing anterogradely labeled fibers after DiI crystal placements in the cingulate cortex. Arrows indicated the cortico-thalamic axon terminal into dorsal thalamus. Asterisks: DiI placed positions. Scale bars; 500 μm.

The following figure supplements are available for figure 6:

**Figure supplement 1.** Loss of corpus callosum in *Nova2-/-* but not *Nova1-/-* mice.

**Figure supplement 2.** Normal formation of anterior commissure axons in *Nova1-/-* and *Nova2-/-* mice.

**Figure supplement 3.** Alternative splicing changes of the genes associated with mouse ACC phenotypes and human ACC syndromes.

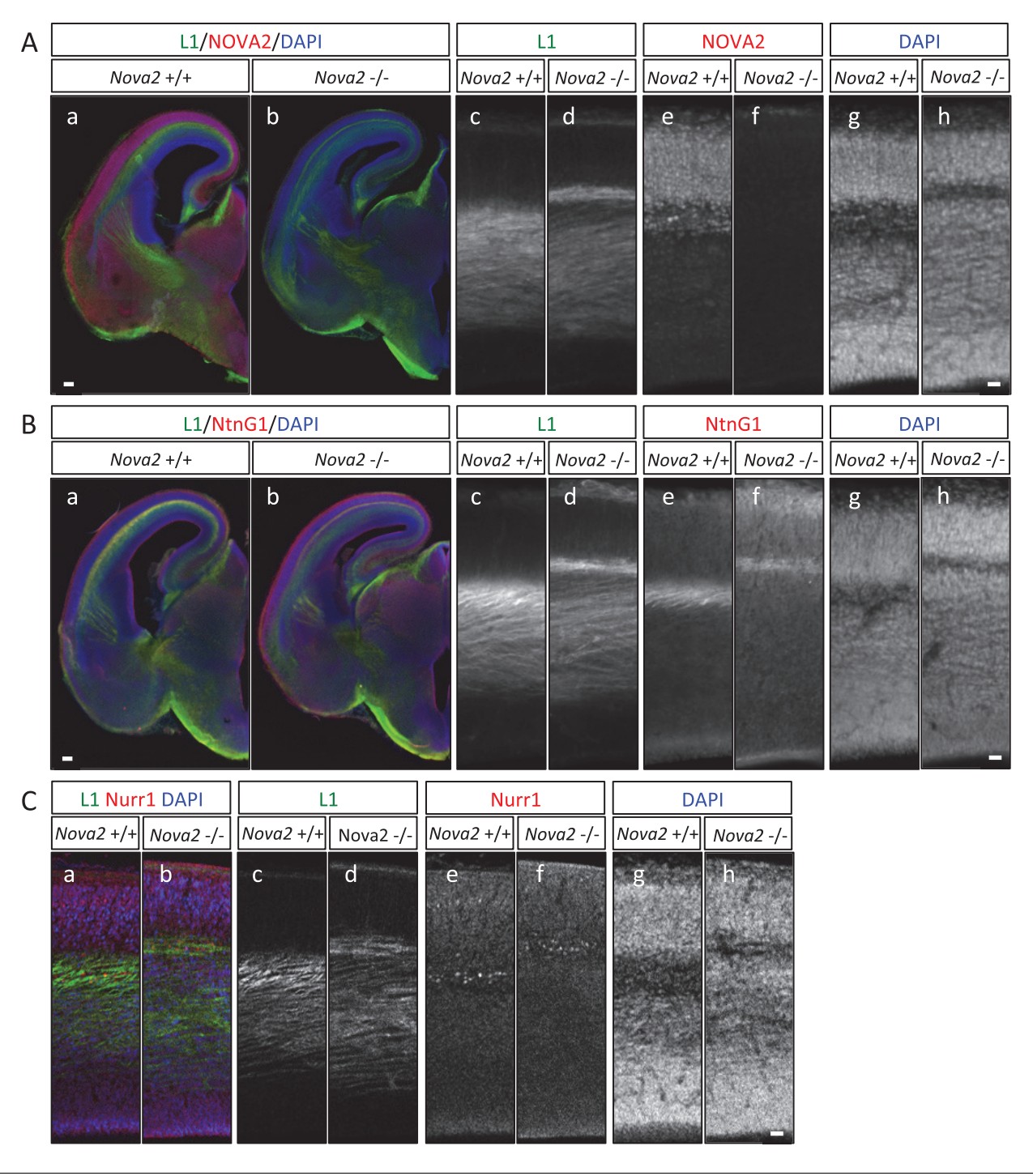

**Figure 7.** Abnormal thalamo-cortical path in the cortex of *Nova2-/-* mice. (**A**) Immunohistochemistry of L1 (c,d) and NOVA2 (e,f) on coronal sections in wild-type (a,c,e,g) and *Nova2-/-* (b,d,f,h) at E16.5. (a,b) Merged coronal section views of L1 (*green*), NOVA2 (*red*), and DAPI (*blue*). (c-h) Higher magnified view of neocortex of (**Aa**) and (**Ab**). Scale bars; 200 μm (a,b), 50 μm (c–h). (**B**) Immunohistochemistry of L1 (c,d) and NTNG1 (e,f) on coronal sections in wild-type (a, c,e,g) and *Nova2-/-* (b,d,f,h) at E16.5. (a,b) Merged coronal section views of L1 (*green*), NTNG1 (*red*), and DAPI (*blue*). (c-h) Higher magnified view of neocortex of (**Ba**) and (**Bb**). Scale bars; 500 μm (a,b), 50 μm (c–h). (**C**) Immunohistochemistry of L1 (c,d) and NURR1 (e,f) in the cortex of wild-type (a,c,e,g) and *Nova2-/-* (b,d,f,h) at E16.5. (a,b) Merged views of L1 (*green*), NURR1 (*red*), and DAPI (*blue*). Scale bar; 50 μm.

E18.5 *Nova2-/-* cortex reverted to patterns seen earlier in E12.5 wild-type cortex. Based on these observations, we conclude that NOVA2 switches the alternative splicing patterns of a series of axon guidance genes during mouse cortical development.

## Axon guidance defects in the brain of *Nova2* but not *Nova1* null mice

The observations that NOVA2 regulates alternative splicing events in transcripts encoding axon guidance signaling factors led us to test whether axon guidance itself is affected in *Nova2* null mice. When coronal and horizontal sections of E18.5 mice brains were double-immunostained for L1, an axonal marker, and TAG-1, a corticofugal axon marker, we discovered that *Nova2-/-* mice, but not wild-type littermates, had agenesis of the corpus callosum (ACC) (14/14 mice) (*Figure 6* and *Figure 6—figure supplement 1A*). ACC was in all *Nova2-/-* mice serial sections examined (*Figure 6—figure supplement 1B*), was seen along with Probst bundles (abnormal collections of cells characteristically seen in patients with ACC; *arrows* in *Figure 6A–o* and *Figure 6—figure supplement 1A–I*), and was also clearly evident in sections stained with the pioneer axon marker Neuropilin-1 (*Figure 6—figure supplement 1C*). Interestingly, *Nova1-/-* mice showed normal CC formation (*Figure 6—figure supplement 1D*). The anterior commissure axons were observed in both *Nova2-/-* and *Nova1-/-* mice (*Figure 6—figure supplement 2*).

To independently corroborate these observations, corpus callosal commissural axons were visualized with DiI anterograde tracer placed into the cingulate cortex. No CC commissural axons were detected that crossed the midline in *Nova2-/-* brain, however, were clearly detected in anterior and posterior sections of wild-type brain at P0 (*Figure 6B*). Corpus callosal commissural axons from the lateral neocortex also failed to cross the midline in *Nova2-/-* mice (*Figure 6—figure supplement 1E*). Interestingly, corticothalamic axons of both wild-type and *Nova2-/-* mice terminated in a similar thalamic region at E18.5 (*arrows* in *Figure 6B*), indicating that only specific aspects of axon guidance are disrupted in *Nova2* null mice. These observations suggest that *Nova2* but not *Nova1* regulate a series of alternative splicing events that are necessary for forming the CC, and demonstrate that NOVA2 and NOVA1 have different functions in CC axon guidance pathways.

To understand the axon guidance phenotype in more detail, we compared the pathway of L1 positive axons traversing the upper cortical layer in the cortex of *Nova2-/-* and wild-type mice at E16.5 and E18.5 (E16.5; *Figure 7* and E18.5; *Figure 6A*). In wild-type mice, L1 positive axons passing between the subplate and subventricular zone were detected as one wide bundle, whereas in *Nova2-/-* mice L1 positive axon routes were separated into two bundles (*Figure 7A*). The L1 positive axons detected in the deeper cortical layer in *Nova2-/-* appeared normal, in that they passed through similar cortical layers as did axons in wild-type mice. In contrast, separated axons passing through the upper L1 positive axonal pathway, which were Netrin-G1 (NTNG1) positive axons, were only seen in *Nova2-/-* mice and were passed along a subplate path enriched in NURR1 (a subplate marker), as revealed by L1/NTNG1 or L1/NURR1 double-immunostaining (*Figure 7B* and *Figure 7C*). These results indicate that only a portion of the L1 positive axonal path, which is NTNG1 positive, is specifically affected by the absence of *Nova2*. We conclude that NOVA2 plays a role as a modifier for a unique set of neuron-specific axons in the developing cortex.

## Motor innervation defects in the ventral diaphragm in NOVA2 dependent manner

We previously reported that NOVA1/NOVA2 regulate the alternative splicing of the *agrin* Z exons (*Ruggiu et al., 2009*). Mice that lack both *Nova* family members fail to cluster AChR at neuromuscular junctions, including those of the diaphragm. Given the role for NOVA2 in axon guidance in the developing cortex (*Figure 6*, *Figure 6—figure supplement 1*, and *Figure 7*), we revisited the role of NOVA1 and NOVA2 on motoneuron axon guidance. The axons of peripheral motor neurons innervating the diaphragm muscle were visualized by immunostaining of neurofilaments at E18.5 (*Figure 8A*) and at E14.5 and E16.5 (*Figure 8B*) and the innervation percentage of muscle in each quadrant of diaphragm at E18.5 was quantified (*Figure 8C*). Although the motor innervation patterns and percentages of the dorsal diaphragm were normal in *Nova2-/-* and *Nova1/Nova2*-double knockout (dKO), those of the ventral diaphragm were severely abnormal with approximately 60–85% of the muscle uninnervated in either *Nova2-/-* and *Nova1/Nova2* dKO mice (p<0.001). Similar abnormalities were observed in *Nova2-/-* and *Nova1/Nova2* dKO mice at E14.5 and E16.5 (*Figure 8B*).

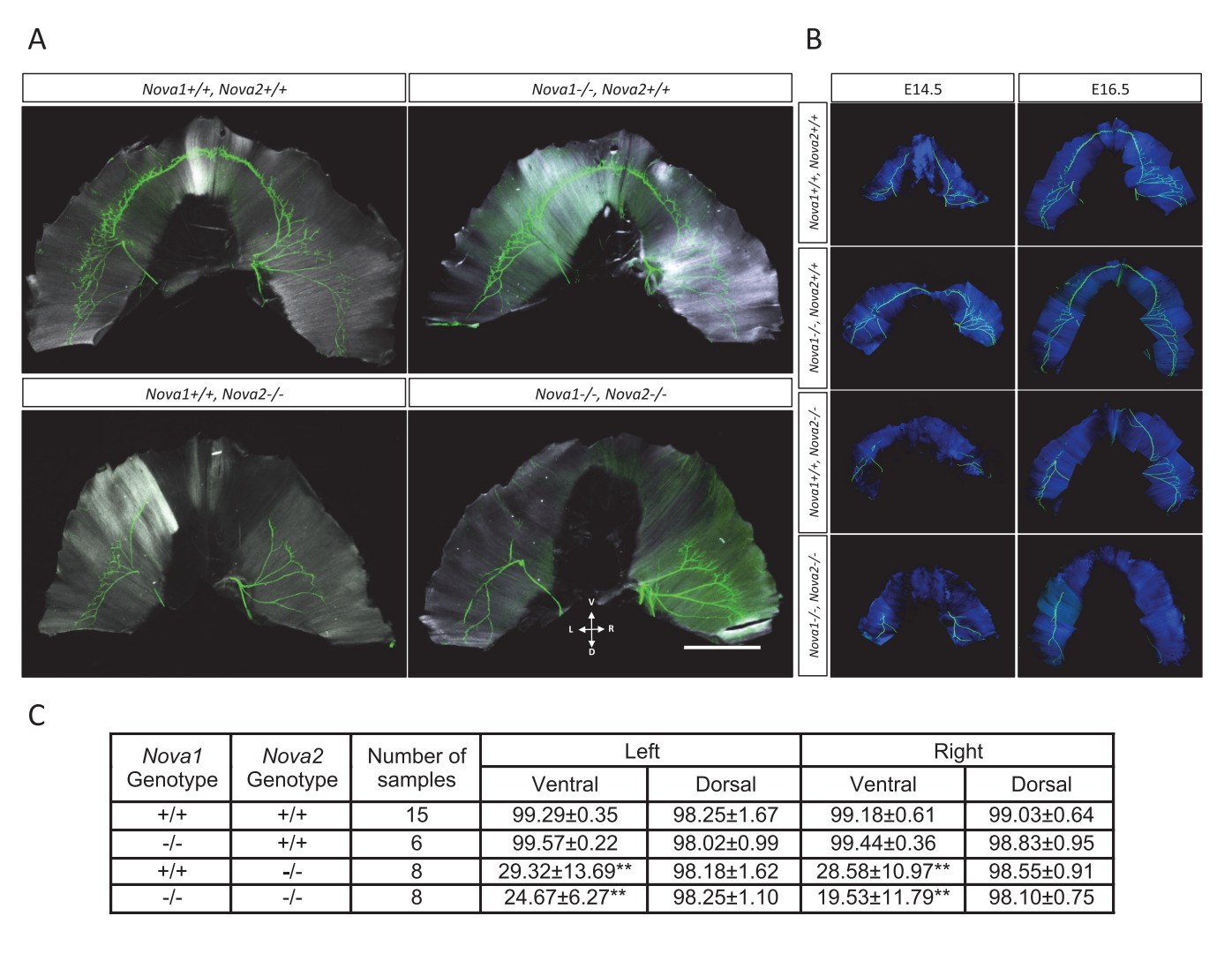

**Figure 8.** Innervation defect in ventral diaphragm of *Nova2-/-* mice. Whole mount staining images of axons (*green*) and muscle (*white* in **A**, *blue* in **B**) in diaphragm of E18.5 (**A**), E14.5 and E16.5 (**B**) wild-type and *Nova1/2* mutants. No differences were observed between left and right diaphragm in each genotypes. Scale: 2 mm. D: Dorsal, L: left, R: right, V: ventral. (**C**) Quantification of innervation percentage of muscle in each quadrant of diaphragm that is covered by the phrenic nerve in E18.5 wild-type and *Nova1/2* mutants. Using the phrenic nerve as reference, measurements of muscle length were taken from the insertion point of the phrenic nerve to the tip of the nerve (X), and from the tip of the nerve to the end of hemidiaphragm (Y), for the ventral and dorsal quadrants of left and right hemidiaphragms. The values indicate the ration of X/(X+Y), expressed as percentage ± standard deviation. **p<0.01 by t-test. Data are presented as mean ± SD.

These innervation defects were specific to *Nova2*-deficient mice, as defects in ventral diaphragmatic innervation were never observed in wild-type or *Nova1-/-* mice. We cannot rule out a role of NOVA1 in the axon guidance of a subset of motoneurons given the slight decrease in ventral innervation in the Nova1/2 double-KO compared to the *Nova2-/-* mice. Nevertheless, taken together, these data demonstrate that in addition to what was observed in CC axons, *Nova2* is also essential for phrenic nerve innervation of the ventral but not dorsal diaphragm.

## Auditory efferent innervation defect in NOVA2 dependent manner

The inner ear is innervated by two types of fibers, afferent innervation consisting of both vestibular and cochlear (spiral ganglion neurons; *Mao et al., 2014*) and from efferent fibers derived from rhombomere 4 of the hindbrain (*Simmons et al., 2011*). Efferents are guided by and extend along

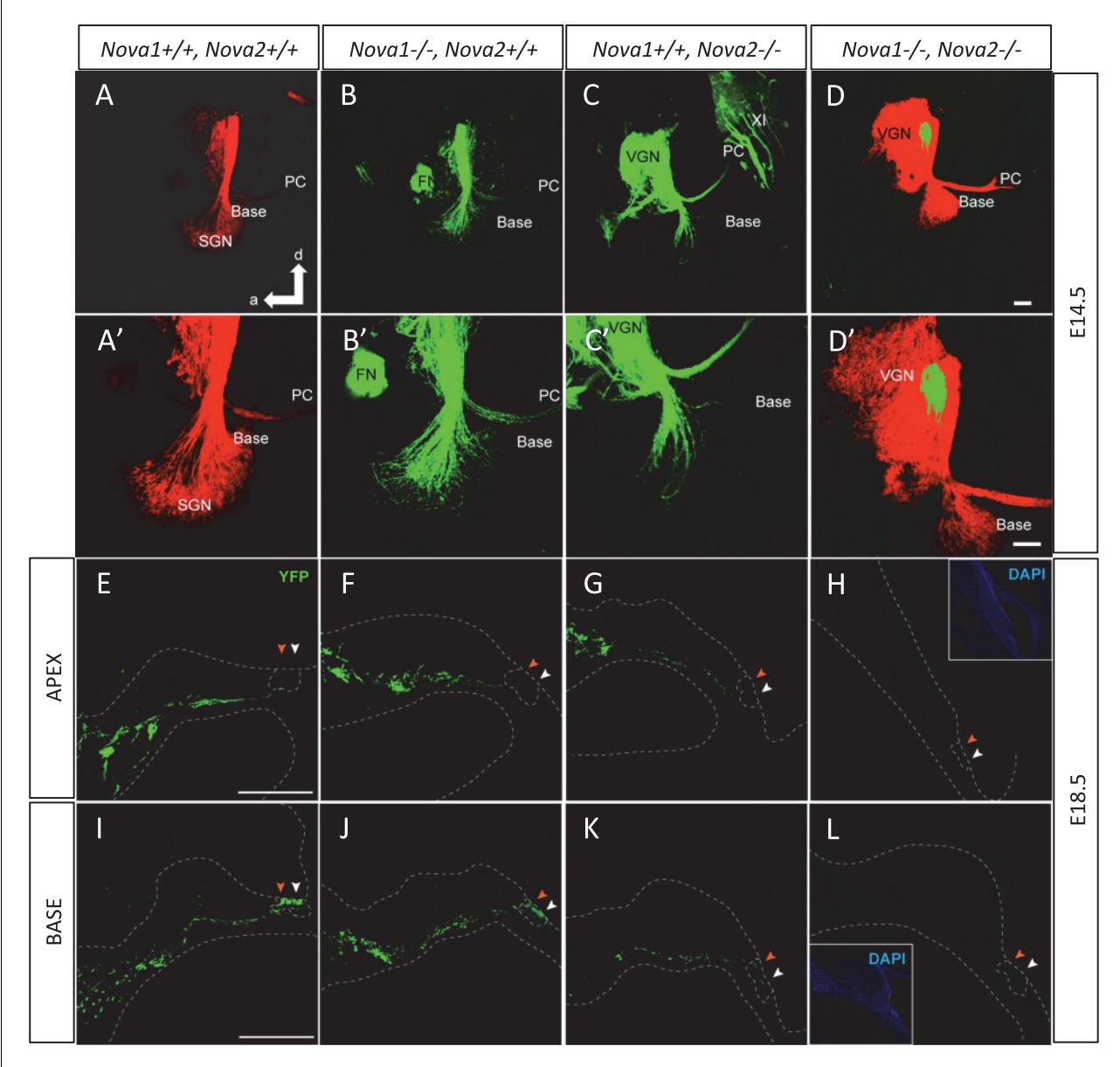

**Figure 9.** NOVA2 expression is necessary for efferent innervation, targeting to the cochlea during embryonic development. (A–D') Olivocochlear and vestibular efferents were labeled with lipophilic dye application to the crossing bundle in rhombomere 4 (*green*) and afferent (*red*) by dye application to the cochlear nuclei. This view shows the left ear viewed from medial. Anterior (a) is to the left and dorsal (d) is up. (A,A') wild-type mice spiral ganglion neurons (SGN) afferents reach the developing organ of Corti at E14.5 in the base and middle turn. Vestibular ganglia (VGN) project to the posterior canal crista (PC) and other vestibular sensory epithelia. (B,B') Dye labeling of efferent fibers to the ear is comparable to wild-type in *Nova1-/-*mice and (C,C') shows reduced fiber growth in *Nova2-/-* mice (D,D'). There is virtually no efferent fiber growth to the ear in *Nova1-/-, Nova2-/-* double knockout mice. Note that in D' the afferent signal was reduced to reveal how far efferents are of the target (E–L). Efferent innervation (*green*) is shown in cochlea cryosections of Nova;YFPJ E18.5 mice. Nuclei are stained with DAPI (*blue*) and actin with Alexa-labeled Phalloidin (*magenta*). Apical (E–H) and basal (I–L) turns are shown. (E,I) wild-type (F,J) and *Nova1-/-* mice show equivalent axonal innervation, (G,K) *Nova2-/-* mice have reduced innervation, (K) especially in the base. (H,L) *Nova1-/-, Nova2-/-* double knockout show no efferent innervation of the cochlea. *Arrowhead* shows approximate location of OHC rows (white) or IHC (orange). Some auto-fluorescence is visible in L. Inset shows nuclear stained (DAPI). Scale bars: 100 μm.

The following figure supplement is available for figure 9:

**Figure supplement 1.** NOVA1 and NOVA2 expression in spiral ganglion (SG) and in superior olive neurons but not in hair cells.

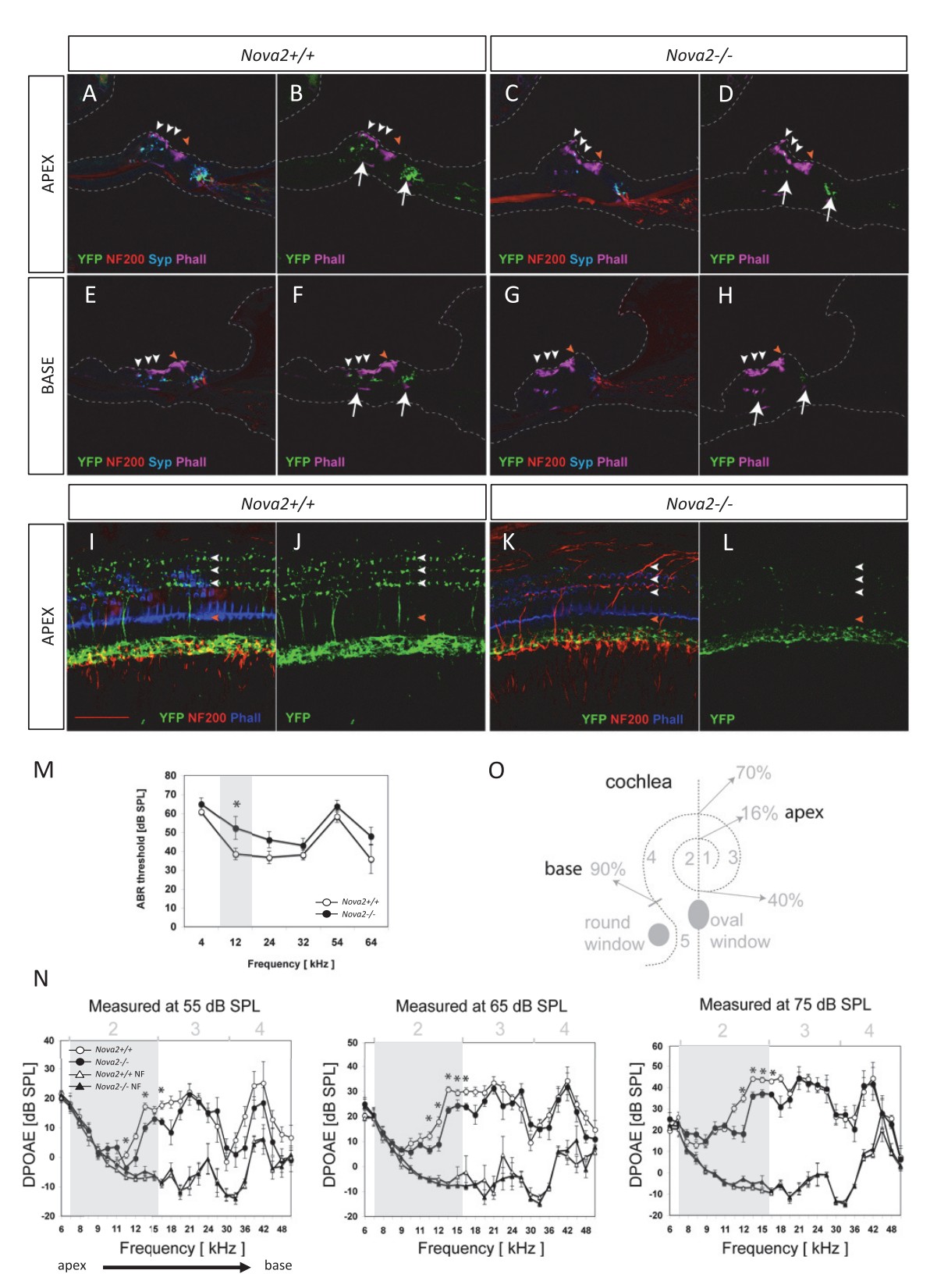

**Figure 10.** Efferents innervation defect and hearing impairment in postnatal Nova2-/- mice. (**A–H**) Cochlea cryosections immunostaining of in *Nova2-/-;*
YFPJand control *Nova2+/+*;YFPJ P14 mice. The images show reduced efferent axons in *green* (YFP, see *arrows*), afferent neurofilaments (NF200) in *red*
*Figure 10 continued on next page*

*Figure 10 continued*

and phalloidin (Phall) labeling of actin-rich hair bundles in *magenta*. The presynaptic marker synaptophysin (syn) in *blue*, shows a reduction of functional efferent innervation. (**A–F**) control mouse, (**C–H**) *Nova2-/-;*YFPJ mouse. (**A–D**) Apical or (**E-H**) Basal organ of Corti. (**I–L**) Cochlea apical turn segment 2 (see below) whole mount preparations immunostaining of (**K, L**) *Nova2-/-;*YFPJ and (**I,J**) *Nova2+/+;*YFPJ P10 mice. The images show reduced efferent axons in *green* (YFP), NF200 in *red* and Phall in *blue*. (**A–L**) *Arrowheads* show approximate location of OHC rows (*white*) or IHC rows (*orange*). (**O**) Representation of the dissection map of the cochlea indicating segments 1–5 and the approximate localization along the length of the cochlea. (**M**) The thresholds of the auditory brainstem response (ABR) to pure-tone stimuli ranging from 10 to 80 dB SPL are increased significantly in *Nova2-/-* mice (n = 4–5 for each frequency) compared to wild-type mice (n = 5–6) at P21-P22. (**N**) The distortion-product otoacoustic emissions (DPOAEs) at $2f_1$-$f_2$ measured at 55, 65 and 75 dB SPL show significant differences between wild-type mice (n = 4) and *Nova2-/-* animals (n = 4) at P21-P22. The noise floor (NF) that was measured simultaneously is also shown for both groups. The approximate acoustic representation of segments 2–4 of the dissected cochlea are shown over each plot and segment 2 is overshadowed in *gray*. Scale bars: 50 μm. *p<0.05.

The following figure supplement is available for figure 10:

**Figure supplement 1.** Reduced efferent innervation and increased afferent innervation to the apex of the *Nova2-/-* mice cochlea.

afferents to reach the sensory epithelia of the ear arriving at approximately the same time (*Fritzsch et al., 1998*; *Ma et al., 2000*). Efferents are a unique population of ventral brainstem neurons derived from facial motoneurons that project to the inner ear in contrast to facial muscle fibers or glands (*Karis et al. 2001*). Some transcription factors have been hypothesized to be relevant for guiding inner ear efferents (*Duncan and Fritzsch, 2013*), however, essentially nothing is known about how these neurons diverge from facial motoneurons to selectively reach the ear to innervate hair cells and afferent processes (*Simmons et al., 2011*). Dye tracing analysis comparing with control mice at E14.5 (*Figure 9A,A'*) showed that *Nova2* (*Figure 9C,C'*) but not *Nova1* (*Figure 9B,B'*) was essential for proper progression of efferent growth along afferents. Moreover, although efferent neurons expressed both NOVA1 (*Figure 9—figure supplement 1I–J*) and NOVA2 (*Figure 9—figure supplement 1F–H*), *Nova1* by itself was not capable of maintaining the function. In the absence of both *Nova1/Nova2,* the efferents stalled when they reached vestibular ganglion neurons (*Figure 9D, D'*) while afferent fibers that also expressed both NOVA1 and NOVA2 (*Figure 9—figure supplement 1A–E, K*) reached the cochlea normally. Our data suggested that *Nova2* played a crucial role for efferent guidance relative to *Nova1,* but both cooperate for normal efferent fiber extension. Interestingly, vestibular efferent innervation which segregated from cochlear efferents at E14.5 (*Bruce et al., 1997*) was unaffected by the absence of *Nova1 or Nova2* alone (see *green* efferents reaching the posterior canal crista (PC) in *Figure 9B,C*) but was also completely stalled in *Nova1/Nova2* double knockouts (*green* label, *Figure 9D*). Postnatal *Nova2-/-* mice vestibular innervation was also normal (not shown).

The cochlea has a tonotopic map where the sensory hair cells at each position along the organ are most sensitive to a particular frequency, forming a continuous gradient from high frequency in the base to low frequency in the apex (*Howard et al., 1988*; *Hudspeth, 1989*; *Liberman, 1982*). Dye tracing analysis showed that the loss of efferent innervation was frequency specific; it was almost completely lost in the base of the *Nova2-/-* cochlea and only partially lost towards the apex (compare *Figure 9A' and C'* ). A more detailed analysis was possible at E18.5 (*Figure 9E–L*), when the medial olivocochlear (MOC) efferent axons expand through the tunnel of *Corti,* prior to their reaching the outer hair cells (OHC) at P2 and establishing functional synapses at P7 (*Simmons et al., 2011*). These axons stained strongly with the Thy-1, YFPJ line that also has high expression in motor neurons, demonstrating the motor origin of auditory efferents. Thy-1 positive axons reached the inner hair cells (IHC) at E18.5 in control (*Figure 9E,I*) and *Nova1-/-* (*Figure 9F,J*) mice but this innervation was reduced in the *Nova2-/-* (*Figure 9G,K*) mice, and this was especially evident in the base (*Figure 9K*). In the absence of both *Nova1/Nova2* there were no labeled axons reaching the cochlea (*Figure 9H,L*), showing that efferents remain stalled. Imaging of efferent innervation at the onset of hearing (P14) showed a clear reduction in axons reaching IHC and specially OHC in the base when comparing *Nova2-/-* mice (*Figure 10C,D,G,H*) to controls (*Figure 10A,B,E,F*). At this age we also found a reduction in efferent synapse formation as demonstrated by reduced labeling with the presynaptic marker synaptophysin (*blue* label at the base of IHC and OHC). Whole mount apical cochlea (Segment 2, see map in *Figure 10O*) at P10 showed a reduction in axons reaching IHC and an

almost complete absence of axons reaching OHC (*Figure 10K,L*) compared to controls (*Figure 10I, J*).

These defects were also evident in apical cholinergic innervation (*Dalian et al., 2001*; *Maison et al., 2003*) after the onset of hearing (P15-P25; *Figure 10—figure supplement 1A–C*). At these stages, we were able to demonstrate that the loss of axonal innervation was functionally significant, as assessed by auditory brainstem responses (ABRs) and distortion products otoacustic emissions (DPOAEs) recordings (*Figure 10M–N*) In addition, an apparent increase in afferent innervation was found in *Nova2-/-* mice by neurofilament immunodetection (*Figure 10A–L* and *Figure 10—figure supplement 1D–E*). The difference was significant in the apex as determined by quantitation of IHC ribbon synapses (*Meyer et al., 2009*) (*Figure 10—figure supplement 1F–J*). Taken together, these data demonstrate a role for NOVA in cochlear innervation, efferent pathfinding, and normal hearing function in vivo.

## Discussion

In the present work, we provide evidence for a novel and unique role for the tissue specific splicing regulator NOVA2 as an axon pathfinding modifier in cortical CC axons, motoneuron, and auditory efferents. In prior studies, integration of transcriptome-wide HITS-CLIP analysis, exon junction microarray data sets, and bioinformatics led to the identification of a large set of NOVA1/2 RNA functional interaction sites and major biological pathways which NOVA1/2 orchestrates in vivo

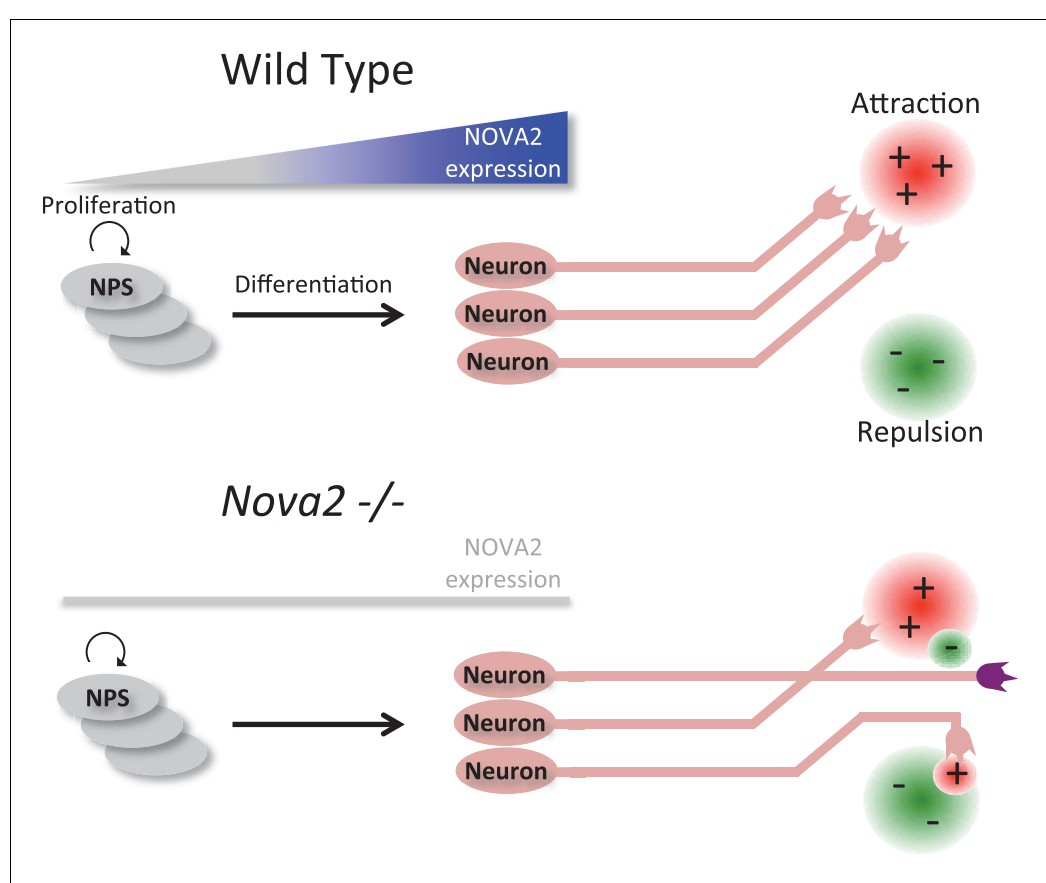

**Figure 11.** Model summarizing a unique role for NOVA2 in the consequence to neuronal axon guidance and outgrowth. The results of HITS-CLIP and RNA-seq analysis combined with histological analysis in *Nova2-/-* mice suggest a model that NOVA2 starts to act on an axon guidance regulatory cascade during neural differentiation in the neurons located on cortical plate. The breakdown of NOVA2-mediated RNA regulation results in the collapse of the normal axon guidance/outgrowth properties in either/both the neuron extending axon or/and the neuron expressing axon guidance cue.

(*Licatalosi et al., 2008*; *Ule et al., 2005b*; *Ule and Darnell, 2006*; *Zhang et al., 2010*). Given identical binding characteristics in shared transcripts (e.g. *Agrin*, others in *Figure 2*), we infer the following: 1) specificity is determined primarily by biochemical interactions: kD of NOVA KH domains binding to accessible YCAY motifs; this might include differential levels of NOVA1/NOVA2 expression within a cell; 2) differential binding is a function of differences in cell expression and/or differences in cell biology (e.g. subcellular distribution of NOVA1 and NOVA2). Evidence for differences in cell expression is clear in the cortex. Differences in actions within cells such as motoneurons that express both NOVA1 and NOVA2 isoforms could arise from different levels of NOVA1 and NOVA2 within the neurons, or due to differential localization of NOVA1 and NOVA2 within the motoneuron, as suggested by the differences in CLIP binding (intron versus 3′ UTR) of the two proteins (*Figure 2F*). The development of single cell-type CLIP would be able to address this issue. Prior genomic and biochemical studies have found no actions that were unique to either NOVA1 or NOVA2 paralogues. The present study identifies NOVA2 as playing a unique role in axonal pathfinding in embryonic development by combining histological analysis with genome wide NOVA1 and NOVA2 specific HITS-CLIP and RNA-seq analysis in *Nova1-/-* and *Nova2-/-* mice. These data build upon previously identified roles for NOVA1 and NOVA2 in early development as modifiers of neuronal migration in late-generated cortical neurons and Purkinje neurons through control of splicing of *Dab1* exon 7b/7c (*Yano et al., 2010*), for *Nova1/Nova2* as essential factors for AChR clustering at the NMJ through the regulation of *Agrin* Z exons (*Ruggiu et al., 2009*), and for *Nova2* in the induction of long term potentiation of slow inhibitory, but not excitatory, post synaptic potential currents after coincidence detection in CA1 hippocampal neurons (*Huang et al., 2005*). The current study provides the first evidence that the formation of corpus callosum in brain, proper innervation of ventral diaphragm, and efferent innervation to the cochlea at developing embryonic stage all require specific actions mediated by NOVA2 (*Figure 11*).

Immunohistochemical and DiI tracer analysis showed that *Nova2* deficiency results in severe ACC (*Figure 6*, and *Figure 6—figure supplement 1*). There are over 50 mouse genes known to be required for the formation of the corpus callosum (reviewed in *Paul et al., 2007*; *Edwards et al., 2014*). NOVA2 HITS-CLIP and RNA-seq analysis revealed that 11 NOVA2 target alternative splicing events on 9 genes (*Dcc, Robo2, Epha5, Slit2, Ank2, Dclk1, Enah, Cask, Ptprs*) intersected with the genes associated with mouse ACC phenotypes and human ACC syndromes (*Figure 6—figure supplement 3*), suggesting that the ACC phenotype in *Nova2-/-* mice may be caused by splicing dysregulation of multiple coordinately NOVA2-regulated transcripts, consistent with previous observations that NOVA proteins regulate subsets of transcripts that mediate coordinate biologic functions (*Ule et al., 2005b*). Given that *Netrin-1* null mice and *Dcc* null mice shows similar ACC phenotype to *Nova2* null mice (*Fazeli et al., 1997*; *Serafini et al., 1996*), we tried to rescue the ACC phenotype with the DCC-long isoform, that is decreased in the cortex of *Nova2-/-* mice, by *in utero* electroporation (data not shown). The expression of DCC-long (long isoform of exon 17) isoform in cingulate cortex at E14.5 was not sufficient to form the commissure corpus callosum, indicating that the DCC-long isoform was not sufficient to rescue the ACC phenotype, and suggesting the possibility that the ACC phenotype in *Nova2-/-* mice either does not involve the DCC-long isoform or is caused by the dysregulation of multiple transcriptomes.

We found that in the cortex, NOVA2 regulates alternative splicing events of two netrin receptors: *Dcc* exon 17 and *Neo1* exon 27 in developmentally regulated manner (*Figure 4*, *Figure 4—figure supplement 1*, and *Figure 5*). In the spinal cord, *Dcc* exon 17 is regulated by NOVA1/2 and required for the spinal commissural neuron development (see accompanying paper by Leggere et al.). Recent structure analysis of netrin-1/NEO1 and netrin-1/DCC-short (short isoform of exon 17) revealed that netrin-1/NEO1 form a 2:2 heterotetrameric netrin-1/NEO1 complex and netrin-1/DCC-short have a continuous netrin-1/DCC assembly (*Xu et al., 2014*). Netrin-1 recognizes the fourth and fifth fibronectin type III domains (termed FN4 and FN5, respectively) of both NEO1 and DCC. Interestingly, the NOVA2-regulated *Dcc* exon 17 is located in the linker region between the FN4 and FN5 domains, and this splicing switch from DCC-short to DCC-long isoform is enough to support the architectural switch of netrin-1/DCC complex from the continuous assembly to the 2:2 assembly (*Xu et al., 2014*). This suggests that NOVA2-dependent splicing regulation results in architecturally distinct netrin-1/DCC complexes that elicit distinct downstream signaling. Changes in NOVA2-dependent regulation of *Dcc* exon 17 splicing during development (*Figure 5*) further indicate that

NOVA2-regulated splicing controls the ability of cortical and spinal neurons to utilize distinct netrin-1/DCC assemblies to respond to a variety of cellular demands.

NEO1, another member of the *Dcc* family, shares ~50% amino acid identity with DCC (*Vielmetter et al., 1994*) and has four known alternative splicing sites. The NEO1 extracellular region is a receptor for repulsive guidance molecule (RGM) family molecules in addition to netrins. Upon ligand binding to the extracellular domain of NEO1 and DCC, their intracellular domains can activate multiple downstream signal transduction pathways, that modulate chemotropic axon guidance (*Bradford et al., 2009*). The inclusion of *Neo1* exon 27 inserts a 53 amino acid sequence into the intracellular region. Inclusion is increased in E18.5 *Nova2-/-* cortex, suggesting that NOVA2-dependent alternative splicing of *Neo1* exon 27 regulates downstream signal transduction. It is well-established that DCC receptors mediate chemo-attraction upon netrin-1 binding, while the UNC5 receptor alone or heterodimerized with DCC elicits chemo-repulsive activity in neuronal tissues (*Hong et al., 1999*). Interestingly, UNC5B interacts with NEO1 as a co-receptor for RGMa and associates with leukemia-associated guanine nucleotide exchange factor (LARG, also termed as ARH-GEF12) to mediate the signal transduction leading to growth cone collapse (*Hata et al., 2009*). Although the abundance of *Arhgef12* RNA in cortex was comparable between E18.5 wild-type and *Nova2-/-* mice (log$_2$FC = 0.10, FDR = 0.78), *Arhgef12* exon 4 inclusion was significantly increased in *Nova2-/-* mice (*Figure 4—figure supplement 1*), suggesting the possibility that NOVA2 modulates RGM signal transduction mediated by NEO1, UNC5b, and ARHGEF12 through coordinate regulation of alternative splicing of those RNAs.

Interestingly, exon 7–8 deletion of *Srrm4/nSR100* gene, an alternative splicing regulator, mice shows partial ACC phenotype and neurite outgrowth defect in diaphragm (*Quesnel-Vallieres et al., 2015*) in addition to hearing defects (see below), indicating that alternative splicing regulation mediated by RBPs (e.g. NOVA2, SRRM4/nSR100) is required for the formation of corpus callosum and neurite outgrowth into diaphragm.

Our analyses were focused on axon guidance related genes but do not rule out the possibilities that axonal pathfinding defect phenotypes in *Nova2-/-* were caused by deficits of multiple biological processes. Several relevant processes were enriched in GO and KEGG pathway analysis, including protein modification, phosphorylation, cellular protein metabolic process, synaptic transmission, chromatin modification, adherens junction, and cell morphogenesis involved in neuron differentiation. Our previous work has showed that the proliferation of cortical neural progenitor cells was not disrupted in *Nova2-/-* (*Yano et al., 2010*). RNA-seq data showed that the transcriptome abundance of the neuronal differentiation markers were comparative between wild-type and *Nova2-/-* mice (*Figure 3—source data 2*), indicating that the differentiation from neuronal progenitor to neuron is comparable between wild-type and *Nova2-/-* mice.

Despite the important role of alternative splicing in the formation of the cochlear tonotopic gradient (*Navaratnam et al., 1997*; *Rosenblatt et al., 1997*; *Miranda-Rottmann et al., 2010*) and in hair cell development and function (*Kollmar et al., 1997*; *Liu et al., 2007*; *Webb et al., 2011*) to date only two alternative splicing factors have been implicated in hearing: *Srrm4/nSR100* (*Nakano et al., 2012*) and *Sfswap* (*Moayedi et al., 2014*). Both are expressed in hair cells necessary for the development of the sensory epithelium, in contrast with *Nova1* and *Nova2*, which are the first neuron-specific splicing factors described to regulate cochlear innervation.

Interestingly auditory efferents are a ventral population of rhombomere 4 facial brachial motoneurons that segregate during development (*Karis et al., 2001*) and while facial innervation is normal, the ventral auditory efferents are selectively affected in the *Nova2-/-* mice (*Figure 9A–D* and data not shown). This specificity is similar to the observation that dorsal motoneuron innervation of the diaphragm is normal while ventral innervation (*Figure 8*) is affected in the *Nova2-/-* mouse, suggesting a more general ventral specialization of axonal guidance regulation by NOVA2. These observations are reminiscent of those seen with defects in GATA-3 innervation of facial motor neurons (*Karis et al., 2001*) and dorsal/ventral pathfinding of limb motoneurons regulated by *EphA4* (*Helmbacher et al., 2000*), suggesting the possibility of an integrated role for NOVA2 in dorsal/ventral axonal pathfinding.

To assess OHC function we measured DPOAEs (*Lonsbury-Martin and Martin 1990*) in *Nova2-/-* mice. There was a significant reduction in DPOAEs in the apex of *Nova2-/-* mice cochlea (Up to 10 dB SPL, *Figure 10N*) where a significant reduction in cholinergic innervation was also found (*Figure 10—figure supplement 1A–C*). Nevertheless the reduction in efferent innervation is

comparatively more severe in the base of the cochlea (*Figure 9* and *10A–H*) suggesting that OHCs sensitive to low frequency sounds are more dependent on *Nova2* controlled efferent innervation. Direct testing of efferent function by electrical activation of these nerves has been shown to reduce the baseline DPOAE intensity (*Liberman et al., 1996*; *Vetter et al., 1999*). Preliminary testing in *Nova2-/-* mice showed that this inhibitory function was not affected (Stéphane S Maison, personal communication), suggesting that the residual efferent innervation was able to sustain the function of further reducing the DPOAEs.

Only an indirect mouse model of efferent innervation defect has been available to date: a mutation in the α9α10 hair cell nicotinic receptor (*Taranda et al., 2009*) in which the DPOAEs threshold levels are reduced by approximately 7–14 dB SPL (a reduction in 10 dB SPL is perceived approximately as half the intensity). In addition, the reduction in DPOAEs found in the *Nova2-/-* mice could explain the ABR threshold elevation at the same frequency range (*Figure 10M*) (*Sun and Kim 1999*; *Taranda et al., 2009*). The opposite possibility, that the defect evidenced by ABRs is responsible for the reduction in DPOAEs is unlikely since it has been reported that even with 80% loss of IHC (with intact OHC) the DPOAEs are only reduced by 10–20 dB SPL (*Trautwein et al., 1996*). It has also been recently shown that afferent innervation to the OHC has no effect on DPOAEs (*Froud et al., 2015*). Despite the reduction in ABRs which measure afferent pathway function, we found an increase in afferent innervation (*Figure 10A–L*) quantified by synaptic ribbon count in *Nova2-/-* mice (*Figure 10—figure supplement 1E–J*) although further study will be necessary to determine if this synapses are functional. Increase in afferent innervation could be a compensatory mechanism to decreased lateral olivocochlear (LOC) efferents, which form synapses on IHC afferents, or a direct effect of the loss of *Nova2* expression in afferent neurons through an ephrin-dependent pathway (*Defourny et al., 2013*). An interesting possibility is that the transient synapses formed between MOC efferents and IHC during development (*Simmons et al., 2011*) have an important role in IHC development that is impaired in *Nova2-/-* mice.

Taken together, our observations suggest that protein/RNA diversity provided by NOVA2-mediated RNA regulation is required for proper axon pathfinding and formation of complex synapses/neural networks, particularly in dorsal/ventral choices, and that alternative splicing switches mediated by NOVA2 may regulate key developmental steps in mammalian biology and pathogenesis of neurological diseases.

## Material and methods

### Generation of *Nova2* null mice

Two fragments from the *Nova2* genomic locus of 2.2 and 6 kb, flanking a 1.5 kb DNA fragment harboring the exon containing the initiating methionine, were cloned into a targeting vector. An *IRES-Cre-FRT-Neo-FRT* targeting cassette was inserted into the *Nova2* locus upstream of the initiator ATG in the first known coding exon of the *Nova2* gene. Linearized targeting vector plasmid was electroporated into ES, and G418-resistant clones (150 ug/mL G418 for 24 hr) harboring homologous recombinants were screened by Southern blot with a 5' *Nova2* genomic probe and injected into mouse blastocysts to produce germline transmitted chimeras. The neomycin cassette was excised by breeding germline-transformed mice with transgenic animals expressing Flp recombinase under the control of the CMV promoter. The original *Nova2+/-* (and *Nova1+/-*) mice obtained in a B57B/6 background were backcrossed for 10 generations to CD-1 and FVB strains and to the thy1-YFP line J (YFPJ) strain (Jackson labs Stock 003709) which has high expression of YFP in motor neurons (*Feng et al., 2000*) or crossed to the R26R strain (Jackson lab stock 003309) for *Nova2* tissue-wide expression analysis. Most experiments were done in CD-1 background, but mice in the inbred FVB background showed no phenotypic differences. FVB mice were used for the hearing tests reported here because they present normal hearing thresholds (*Zheng et al., 1999*) and less inter individual differences. All procedures in mice were performed in compliance with protocols approved by the Institutional Animal Care and Use Committee (IACUC) of the Rockefeller University or the Comité de déontologie de l'expérimentation sur les animaux (CDEA) of the Univeristy of Montreal.

## Antibodies

Primary antibodies used for immunohistochemistory and western blot were as follows; goat anti-NOVA2 (C-16) (sc-10546, Santa Cruz), rabbit anti-NOVA1 [EPR13847] (ab183024, abcam), human anti-pan NOVA (anti-Nova paraneoplastic human serum), rabbit anti-PTBP2 (*Polydorides et al., 2000*), rat anti-L1 (MAB5272, Millipore), goat anti-TAG1/Contactin-2 (AF4439, R&D systems), goat anti-NetrinG1a (AF1166, R&D systems), rabbit anti-NURR1 (M-196) (sc-5568, Santa Cruz), rabbit anti-neurofilament (AB1981, Chemicon), mouse anti-NF200 (N5389, SIGMA), mouse anti-ctbp2 (612044, BD), rabbit anti-myoVI (sc-50461, Santa Cruz), and goat anti-neuropilin-1 (AF566, R&D systems). Anti-NOVA1 and anit-NOVA2 antibody specificity for immunohistochemistory and immunoprecipitation was confirmed (*Figure 1—figure supplement 2*).

## Immunohistochemistory

E16.5, E18.5 mice brains were fixed with 4% praraformaldehyde (PFA)/PBS at 4 degrees overnight, and sequentially replaced to 15% sucrose/PBS and 30% sucrose/PBS at 4 degrees for cryo-protection, then embedded in OCT compound. Frozen brains were sliced into 80 μm thick sections on a cryostat (CM3050S, LEICA). Brain sections and whole embryos fixed with 4% PFA/PBS were subjected to floating or whole-mount immunohistochemistry. These samples were washed three times with PBS at room temperature (RT), incubated with 0.2% Triton X-100/PBS at R.T., blocked with 1.5% normal donkey serum (NDS)/PBS at RT, and then incubated overnight at 4 degrees with primary antibodies or Alexa-dye conjugated phalloidin in 1.5% NDS/PBS followed by incubation with Alexa 488, 555, or 647 conjugated donkey secondary antibodies (1:1000) in 1.5% NDS/PBS. Images of immunostained specimens were collected by BZ-X700 (KEYENCE), fluoscence microscopes Axioplan 2 (Zeiss), or an inverted TCS SP8 laser scanning confocal microscope (LEICA) at The Rockefeller University Bio-Imaging Resource Center.

## Cochlea tissue processing and imaging

Embryos were obtained by C-section of time-mated dams and fixed by overnight immersion in 4% PFA. Postnatal mice were intracardialy perfused with 4% PFA, cochleae immediately removed and perfused through the round window with a 1 ml syringe. After post-fixation 2:30 hr at 4°C cochleae were washed in PBS, directly dehydrated in 30% sucrose at 4°C or previously decalcified for 24–48 hr in 120 mM EDTA in PBS. For beta-galactosidase expression analysis the decalcified cochleae were perfused through the round window with M-1 embedding matrix (Thermo Scientific) and frozen in a block of the same matrix, 40 μm cryosections were incubated in staining solution (4 mM potassium ferricyanide, 4 mM potassium ferrocyanide, 1 mg/ml x-gal) dissolved in wash buffer (2 mM magnesium chloride, 0,01% sodium deoxycholate, 0,1% NP40 and 0.2 mM buffer phosphate pH 8.0) overnight at 37°C, washed in wash buffer 3 hr at 4°C, post-fixed again in 4% PFA and washed in PBS before mounting. For AChE staining the non-decalcified cochleae were dissected by carefully chipping the bone away and incubated in a solution containing 0.5 mg/ml acetylcholine iodide as explained elsewhere (*Willott, 2001*). For whole mount immunostainings the decalcified cochleae were dissected cutting into half turns (See *Figure 10O*), permeabilized and blocked in 0.1% TX100, 5% donkey serum in PBS. Primary and secondary antibodies were incubated sequentially 24 hr at 4°C. Washes were done in 0.05% TX100. For cryosections the decalcified cochleae perfused through the round window with M-1 embedding matrix (Thermo Scientific) or the fixed embryos were cut into 14–20 μm sections, permeabilized and blocked in 0.1% Triton-X100, 5% donkey serum in PBS. Primary antibodies were incubated 24 hr at 4°C and secondary antibodies for 2 hr at room temperature. Washes were done in PBS. Alexa labeled Phalloidin (1:100) was added to the secondary antibody mix. Images were collected using a Zeiss LSM 510 confocal microscope at The Rockefeller University Bio-Imaging Resource Center and a Zeiss LSM 700 confocal microscope at the IRIC core of the University of Montreal.

## Axon tracing

For DiI trace analysis, DiIC18(3) crystal (D3911, molecular probes) were placed in cingulate cortex or lateral neocortex of P0 brain. After 8–21 days at 37 degrees in 4% PFA/PBS to allow dye diffusion, samples were cryo-protected with 30% sucrose/PBS at 4 degrees, embedded in OCT compound, and cut into 100 μm sections on cryostat. Counterstaining was with 1 μg/mL DAPI (4',6-Diamidino-2-

phenylindole dihydrochloride). The images of immunostained specimens were collected by a fluo-scence microscope Axioplan 2 (Zeiss). Inner ear afferents and efferents were labeled with lipophylic (NeuroVue) dyes (Molecular Targeting Technologies; MTTI). These dyes were placed into specific areas for selective labeling of nerve fibers (*Fritzsch et al., 2005*; *Duncan et al., 2015*) Dyes were placed into the cerebellum and the alar plate of rhombomere 2 to label the afferent fibers to the ear. Dye was also placed into the basal plate of rhombomere 4 for efferent nerve labeling. The prep-arations were then incubated in 4% PFA at 36 degrees C for three days. After dye diffusion the ear was dissected out and placed on a glass slide with glycerol and a cover slip on top to be imaged with a Leica SP5 confocal microscope.

## Western blot and semi-quantitative RT-PCR

Western blotting and RT-PCR analyses were performed as described elsewhere (*Yano et al., 2010*). For quantification, three biological replicates were used and quantified from obtained images by using ImageJ free software (http://rsb.info.nih.gov/ij/index.html).

## Hearing tests

ABRs were recorded from deeply anesthetized mice. The positive needle electrode was inserted subdermally at the vertex, the negative electrode was placed beneath the pinna of the left ear, and the ground electrode was located on the hind leg. ABRs were evoked by tone bursts ranging from 4–64 kHz and were produced by a closed-field electrostatic speaker connected to a driver (EC1 and ED1; Tucker-Davis Technologies). The 5 ms tone bursts were presented 33.3 times per second; their 0.5 ms onsets and offsets were tapered with a squared cosine function. The speaker's audio output was transmitted into the ear through a custom acoustic assembly. Sound-pressure levels were mea-sured with a calibrated microphone and preamplifier connected to a conditioning amplifier (4939A011 and 2690A0S1, Brüel and Kjær). The response was amplified x10,000 and bandpass fil-tered at 0.33 kHz (P55, Natus Neurology Inc.). The amplified response was then digitally sampled at 10 µs intervals with a data acquisition device (PCIe-6353, National Instruments) controlled by custom software (LabVIEW 2010, National Instruments). The responses to 1000 bursts were averaged at each intensity level to determine the threshold, defined as the lowest level at which a response peak was distinctly and reproducibly present. For each frequency, sound-pressure level was decreased from 80 dB SPL in 5 dB steps until threshold was reached. DPOAEs were then elicited with an acous-tic coupler that allowed for recording ear-canal sound pressure levels via a probe tube concentrically situated within the common sound-delivery tube. The acoustic coupler consisted of two electrostatic speakers (EC1, Tucker-Davis Technologies) to generate primary tones and a miniature microphone (EK23103, Knowles) to measure ear-canal sound pressure (coupler from Mike Ravicz, EPL, Eye&Ear inf, Boston). The speakers and the microphone were both calibrated using the calibrated micro-phone described above. The $2f_1 - f_2$ distortion product was measured with $f_2$ = 6–54 kHz, $f_2/f_1$ = 1.2, and $L_1 = L_2$ = 5575 dB SPL. The acoustic signal was amplified by a preamplifier (ER10B+, Ety-motic Research) and the sound pressure measured in the ear canal was digitally sampled at 10 µs intervals with the data-acquisition system described above. Each frequency pair was presented for 1 s. After fast Fourier transforms had been computed and averaged over ten consecutive traces, the amplitudes of the $2f_1 - f_2$ distortion product and the surrounding noise floor (+/- 100 Hz of $2f_1 - f_2$) for each frequency pair were determined, a procedure requiring 17 s of data acquisition and proc-essing time.

## HITS-CLIP

NOVA1 and NOVA2 CLIP was performed on E18.5 wild-type cortex using three biological replicates, as described elsewhere (*Licatalosi et al., 2008*). High-throughput sequencing was performed at the Rockefeller University Genome Resource Center. Sequence tags were aligned to the mouse genome (mm9) by novoalign. Unique tags were collected by eliminating PCR duplicates.

## RNA-seq and analysis

E18.5 mouse cortex RNA from wild-type and *Nova2-/-* littermate and E18.5 mouse cortex and mid-brain/hindbrain RNA from wild-type and *Nova1-/-* littermate (three biological replicates for each genotype) was prepared using Trizol (ambion) and ribosomal RNA was removed from 1 µg RNA

using Ribo-Zero rRNA removal Kit (epicentre). Standard RNA-seq libraries were prepared using Tru-Seq RNA Sample Preparation Kit v2 (illumina) following manufacturer's instructions. High-throughput sequencing was performed on Hi-seq 2500 (illumina) to obtain 125 nucleotide paired-end reads at New York Genome Center. Reads were aligned to the mouse genome (mm10) using OLego (*Wu et al., 2013*). The downstream splicing analysis was performed as described previously (*Yan et al., 2015*) and then liftovered to mm9. Differential expression analysis was performed with the R package edgeR (*Robinson et al., 2009*) available on Bioconductor 3.0 (*Gentleman et al., 2004*).

## Accession numbers

RNA-seq and HITS-CLIP data have been deposited in GEO under accession number GSE69711 (http://www.ncbi.nlm.nih.gov/geo/query/acc.cgi?acc=GSE69711).

## Acknowledgements

We thank Claudia Scheckel, Hun-Way Hwang, Michael J Moore, and Jennifer C Darnell for critical discussion and reading of manuscript and Shengdong Ke for help with bioinformatics processing of HITS-CLIP and RNA-seq data sets, and are grateful to all other members of Robert B Darnell's laboratory for kind support. We thank M Charles Liberman and the members of his laboratory for technical help and valuable discussion. This work was supported by grants from the National Institutes of Health (NS034389 and NS081706) to RBD. SM's research was supported in part by a grant of the Hearing Health Foundation and by CIHR grant RNI00109 to RA. ZC was supported by the Boettcher Foundation. HJJ was supported by the NIH (R01EY024261). YS was supported by a JSPS postdoctoral fellowship for research abroad and The Naito Foundation Subsidy for Inter-Institute Researches. BF and JAD were supported by P30 DC010362 and the Office of Vice President for Research at the University of Iowa. RBD and AJH are Investigators of the Howard Hughes Medical Institute.

## Additional information

### Funding

| Funder | Grant reference number | Author |
|---|---|---|
| Japan Society for the Promotion of Science | JSPS postdoctoral fellowship for research abroad | Yuhki Saito |
| Naito Foundation | Subsidy for Inter-Institute Researches | Yuhki Saito |
| Hearing Health Foundation | | Soledad Miranda-Rottmann |
| National Institutes of Health | R01EY024261 | Harald J Junge |
| Boettcher Foundation | | Zhe Chen |
| Canadian Institutes of Health Research | RNI00109 | Roberto Araya |
| Howard Hughes Medical Institute | | A J Hudspeth Robert B Darnell |
| National Institutes of Health | R01 NS069473 | Robert B Darnell |
| National Institutes of Health | R01 NS081706 | Robert B Darnell |
| National Institutes of Health | R01 NS034389 | Robert B Darnell |
| Office of Vice President for Research | P30 DC010362 | Jeremy S Duncan Bernd Fritzsch |

The funders had no role in study design, data collection and interpretation, or the decision to submit the work for publication.

## Author contributions

YS, Designed experiments, Prepared RNA-seq library and performed RT-PCR, Performed all other experiments and data analysis, Wrote the manuscript; SM-R, Designed experiments, Performed hearing research in the Nova mutant mice, Finished experiments in RA's laboratory, Wrote the manuscript, Analysis and interpretation of data; MR, Designed experiments, Characterized the Nova2-/- mice general phenotype and the spinal cord motoneuron defect, Wrote the manuscript, Acquisition of data, Analysis and interpretation of data; CYP, Performed alternative splicing analysis; JJF, Prepared RNA-seq library and performed RT-PCR, Acquisition of data; RZ, Helped design and construct the Nova2-/- mice, Acquisition of data; JSD, Performed and analyzed auditory dye tracing experiments; BAF, Built the auditory function setup and programed custom software, Acquisition of data, Analysis and interpretation of data; HJJ, ZC, Wrote the manuscript, Drafting or revising the article; RA, Contributed unpublished essential data or reagents; BF, Performed hearing research in the Nova mutant mice, Wrote the manuscript, Conception and design, Contributed unpublished essential data or reagents; AJH, Performed hearing research in the Nova mutant mice, Conception and design, Contributed unpublished essential data or reagents; RBD, Supervised the project, Wrote the manuscript, Conception and design

## Author ORCIDs

Jeremy S Duncan, http://orcid.org/0000-0002-5555-3273
Zhe Chen, http://orcid.org/0000-0003-0683-9491
A J Hudspeth, http://orcid.org/0000-0002-0295-1323
Robert B Darnell, http://orcid.org/0000-0002-5134-8088

## Ethics

Animal experimentation: This studies were performed in compliance with protocols (#14678 and #07069) approved by the Institutional Animal Care and Use Committee (IACUC) of the Rockefeller University or with protocols (13-185 and 15-002) approved by the Comité de déontologie de l'expérimentation sur les animaux (CDEA) of the Univeristy of Montreal.

# Additional files

## Major datasets

The following dataset was generated:

| Author(s) | Year | Dataset title | Dataset URL | Database, license, and accessibility information |
|---|---|---|---|---|
| Yuhki Saito | 2016 | Nova HITS-CLIP and RNA-Seq in mouse cortex | http://www.ncbi.nlm.nih.gov/geo/query/acc.cgi?&acc=GSE69711 | Publicly available at NCBI Gene Expression Omnibus (accession no: GSE69711) |

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
