## [Decision Letter]

Thank you for submitting your article "NOVA2-mediated RNA regulatory networks are required for axonal pathfinding during development" for consideration by *eLife*. Your paper has been reviewed by three reviewers, one of whom is a member of the Board of Reviewing Editors, and it was also evaluated by a Senior Editor.

The reviewers have discussed the reviews with one another and the Reviewing Editor has drafted this decision to help you prepare a revised submission.

All three of the reviewers agree that the strength of the paper lies in finding that Nova2 uniquely regulates alternative splicing of axon guidance-related genes in cortical development, that the Nova2 null mice have interesting axonal defects in cortex, spinal cord, and brain stem. All also agree that the quality of the molecular data is excellent.

The major limitations are: 1) there is no causal link between any of the splicing defects and the phenotypes, including agenesis of the Corpus Callosum; 2) the authors do not even attempt to relate their molecular findings to the over 60 genes already known to play a role in development of the Corpus Callosum; they do not credit some important relevant papers and do not look at other potential candidates that emerged from their GO analysis; and lastly there are no data as to how the alternatively spliced variants differ in activity.

While the reviewers see these limitations, they have decided that together with the Chen et al. paper, the two manuscripts report on related and biological observations.

That said there are several things outlined below that we do hope you can address in a revision. The full reviews are also included at the end of this summary.

The issues that must be addressed:

1) The loss of the corpus callosum is a severe and convincing phenotype in *Nova2* mutants. The authors focus their discussion of this phenotype almost entirely on the netrin-1/DCC signaling pathway for the good reason that Nova2 regulates the alternative splicing of two netrin receptors, DCC and Neo1. However, they do not test this directly. The link between the splicing target and the phenotype would be significantly strengthened by the addition of data testing whether DCC or Neo1 can rescue the ACC phenotype. These would be challenging but valuable experiments.

Without the DCC or Neo1 rescue experiments, the authors should widen the discussion about the possible targets affected in the Nova2A mutant that could give rise to the agenesis phenotype. There are around 60 mouse genes known to be required for the formation of the corpus callosum (e.g. Paul et al. 2007; Donahoo and Richards, 2012) so it would valuable to try to establish how many of these identified genes are targets of Nova2. How many of these 60 genes are present in the Nova2A target data? Loss of Neuropilin1, for example, gives rise to ACC (Lim et al. 2015) and the EphB1 and EphB2 intracellular signalling domains are required for the formation of the CC (Robichaux et al. 2015). These two studies also report the loss of the anterior commissure in addition to the CC. Does the *Nova2* mutant show a defect in axons crossing the AC?

2) Overall, the authors do not fully credit highly relevant studies in the literature. In addition to examples above, Blencowe's group published a particularly relevant study with remarkably similar findings to the present work last year that is not cited here (Quesnel-Vallieres et al., G & D 2015). They report that the loss of nSR100, an alternative splicing factor in neurons, gives rise to CC agenesis and to phrenic nerve defects. In addition, a mutation in nSR100 has been reported to cause hearing defects and should probably be cited along with the other two the authors cite in the statement "to date only two alternative splicing factors have been implicated in hearing: *Srrm4* (Nakano et al. 2012) and *Sfswap* (Moayedi et al. 2014)".

3) Given the limitations of the Study, perhaps a more accurate title would be "NOVA-2 mediated alternative splicing is required for axon pathfinding during development." Indeed, the data certainly demonstrate that this is the case. While "regulatory networks" allude to profound insights, to what extend this is true remains unclear, while it is clear that "alternative splicing" is important.

4) The authors should provide schematic representations for the developmental systems to make the data more broadly accessible to readers. It would also be helpful to make it clear to the reader that the specific phenotypes are complex and as NOVA-2 is broadly expressed how specific developmental defects arise may be a consequence not only of changes in transcripts in a specific cell type, but also changes in transcripts in different ways in different neurons. That is, the phenotypic analysis assumes cell autonomy and no data addresses this issue in the paper.

5) Furthermore, whether these axon guidance defects arise solely from alteration of transcripts involved in axon guidance or indirectly by disrupting the regulation of other genes involved in neuronal differentiation is unclear. That said, at least some guidance receptors are targets for NOVA-2 regulation and this observation as well as observations in the companion paper provide support for a direct role in this process. It is important to note, however, that the GO analysis presented in the supplementary material [Supplementary-material SD1-data] indicates that other categories of genes are also significantly altered in NOVA-2 mutants, including genes involved protein modification, phosphorlyation, metabolism and chromatin modification. Thus, to what extent the observed phenotypes reflect alterations in axon guidance receptors per se, as opposed to proteins acting to regulate neuronal differentiation more generally and axon guidance only indirectly remains unclear. All this needs to be addressed in the Discussion.

6) The significance of the changes in axon guidance receptors is discussed in the context of the long and short forms of DCC, the netrin receptor. This discussion really belongs in the other manuscript by Chen et al., and that focuses specifically on the NOVA-2 regulation of this pathway.

*Reviewer #1:*

The manuscript investigates the functional differences between the two Nova isoforms, Nova1 and Nova2. The authors compare transcriptome-wide HITS-CLIP and RNA-seq data from *Nova1* and *Nova2* mutants and report evidence showing that Nova2 uniquely regulates the alternative splicing of axon guidance-related genes in cortical development. They also report that loss of Nova2, but not Nova1, gives rise to axonal defects in the cortex, spinal cord and brainstem, particularly in ventral motoneuron axons. Overall, the results indicate that Nova2 controls alternative splicing of some key components involved in axon guidance.

This is an interesting study that compares Nova1 and Nova2 target transcripts and function. The study follows up on previous work from Darnell and colleagues and provides novel evidence for distinct isoform-specific targets and function. The transcriptome-wide analyses are robust and compelling. The functional axonal defects seen with loss of Nova2A are very clear and well described. However, the study does not identify the Nova2A targets that underlie the axon guidance defects. This weakness could potentially be addressed by further experiments.

1) The loss of the corpus callosum is a severe and convincing phenotype in *Nova2* mutants. The authors focus their discussion of this phenotype almost entirely on the netrin-1/DCC signaling pathway for the good reason that Nova2 regulates the alternative splicing of two netrin receptors, DCC and Neo1. However, they do not test this directly. The link between the splicing target and the phenotype would be significantly strengthened by the addition of data testing whether DCC or Neo1 can rescue the ACC phenotype. These would be challenging but valuable experiments.

Without the DCC or Neo1 rescue experiments, the authors should widen the discussion about the possible targets affected in the Nova2A mutant that could give rise to the agenesis phenotype. There are around 60 mouse genes known to be required for the formation of the corpus callosum (e.g. Paul et al. 2007; Donahoo and Richards, 2012) so it would valuable to try to establish how many of these identified genes are targets of Nova2. How many of these 60 genes are present in the Nova2A target data? Loss of Neuropilin1, for example, gives rise to ACC (Lim et al. 2015) and the EphB1 and EphB2 intracellular signalling domains are required for the formation of the CC (Robichaux et al. 2015). These two studies also report the loss of the anterior commissure in addition to the CC. Does the *Nova2* mutant show a defect in axons crossing the AC?

2) Overall, the authors do not fully credit highly relevant studies in the literature. In addition to examples above, Blencowe's group published a particularly relevant study with remarkably similar findings to the present work last year that is not cited here (Quesnel-Vallieres et al., G & D 2015). They report that the loss of nSR100, an alternative splicing factor in neurons, gives rise to CC agenesis and to phrenic nerve defects. In addition, a mutation in nSR100 has been reported to cause hearing defects and should probably be cited along with the other two the authors cite in the statement "to date only two alternative splicing factors have been implicated in hearing: *Srrm4* (Nakano et al. 2012) and *Sfswap* (Moayedi et al. 2014)".

*Reviewer #2:*

This is a provocative paper from Darnell and colleagues. Using a combination of HITS-CLIP and RNA-sequencing, transcripts encoding proteins regulating axon guidance were identified as targets preferentially regulated by the NOVA-2 splicing factor (selectively when compared to NOVA-1). Additional data are presented in support of specific alternative splicing events selectively regulating axon guidance receptors via a NOVA-2 dependent mechanism. In the second part of the paper, genetic studies are presented demonstrating defects in axon guidance in different developmental contexts. The authors argue that these data support the conclusion as stated in their title "NOVA-1 mediated RNA regulatory networks are required for axonal pathfinding during development". Perhaps a more accurate title would be "NOVA-2 mediated alternative splicing is required for axon pathfinding during development." Indeed, the data certainly demonstrate that this is the case. While "regulatory networks" allude to profound insights, to what extend this is true remains unclear, while it is clear that "alternative splicing" is important.

The RNA-sequencing and HITS clip data are impressive and comprehensive and reveal an impressive array of transcripts selectively regulated by NOVA-2. Together this makes a compelling argument that NOVA-2 broadly sculpts the neuronal proteome. The genetic data establish marked axon guidance defects in several different developmental contexts and thus broadly supports the notion of a critical role for NOVA-2 in regulating axon guidance. These data are compelling. My only suggestion here is that the authors provide schematic representations for the developmental systems to make the data more broadly accessible to readers. It would also be helpful to make it clear to the reader that the specific phenotypes are complex and as NOVA-2 is broadly expressed how specific developmental defects arise may be a consequence not only of changes in transcripts in a specific cell type, but also changes in transcripts in different ways in different neurons. That is, the phenotypic analysis assumes cell autonomy and no data addresses this issue in the paper.

Furthermore, whether these axon guidance defects arise solely from alteration of transcripts involved in axon guidance or indirectly by disrupting the regulating of other genes involved in neuronal differentiation is unclear. That said, at least some guidance receptors are targets for NOVA-2 regulation and this observation as well as observations in the companion paper provide support for a direct role in this process. It is important to note, however, that the GO analysis presented in the supplementary material [Supplementary-material SD1-data] indicates that other categories of genes are also significantly altered in NOVA-2 mutants, including genes involved protein modification, phosphorlyation, metabolism and chromatin modification. Thus, to what extent the observed phenotypes reflect alterations in axon guidance receptors per se, as opposed to proteins acting to regulate neuronal differentiation more generally and axon guidance only indirectly remains unclear.

The significance of the changes in axon guidance receptors is discussed in the context of the long and short forms of DCC, the netrin receptor. This discussion really belongs in the other manuscript that focuses specifically on the NOVA-2 regulation of this pathway.

In general, I feel that the two papers together certainly are greater than the sum of the two. I suggest that authors play down the notion of a network here as it says very little and implies a level of understanding that is by no means achieved here. Perhaps, the authors could move the discussion of DCC from this manuscript to the other one, and use the space to address the question of how NOVA-2 regulates neural development, and perhaps expand this to explore the relationship to other global splicing regulators of neuronal differentiation such RbFox.

*Reviewer #3:*

This work provides an extensive analysis of loss-of-function phenotypes for NOVA1 and NOVA2 mutant mice. Using a combination of genome-wide approaches the authors identify an alternative splicing program regulated by the NOVA RNA-binding proteins. This analysis extends previously reported data sets on such targets.

Amongst many other targets, there is a significant de-regulation of several axon guidance regulators, in particular components that have been shown to contribute to growth and guidance at the midline. In parallel, the authors report a series of axon extension and guidance phenotypes in the mutant mice.

The datasets are extensive and rigorous. What is a weakness of the work is the lack of data demonstrating a causal link between any of the splicing targets and the anatomical phenotypes.

There is also no insight into how the alternative splice isoforms of the guidance receptors may differ in activity.

The main contributions of the work are the description of novel anatomical phenotypes and splicing targets, as well as the identification of distinct phenotypes in NOVA2 versus NOVA1 mutant mice. However, the mechanism underlying this apparent specificity remains unclear (cell type-specificity of NOVA isoform expression or unique interactions with co-factors or RNA elements?).

The data is clear, analysis rigorous, and the interpretation of the results is appropriate. Adding more mechanism would mean a substantial amount of experimental work. In the present form this appears to me like a borderline case for publication in *eLife*.

---

## [Author Response]

The issues that must be addressed:

1) The loss of the corpus callosum is a severe and convincing phenotype in Nova2 mutants. The authors focus their discussion of this phenotype almost entirely on the netrin-1/DCC signaling pathway for the good reason that Nova2 regulates the alternative splicing of two netrin receptors, DCC and Neo1. However, they do not test this directly. The link between the splicing target and the phenotype would be significantly strengthened by the addition of data testing whether DCC or Neo1 can rescue the ACC phenotype. These would be challenging but valuable experiments.

We appreciate the reviewers’ suggestion, and have tried this challenging experiment. Since it has not been reported that *Neo1* KO mice have ACC phenotype, we tried to rescue the ACC phenotype with the DCC-‐long isoform that is decreased in the cortex of *Nova2*-‐KO mice. We undertook in uteroelectroporation of plasmids expressing the DCC-‐long isoform. Plasmid expressing both the *DCC* long isoform and DsRed as a positive control were electroporated into the cingulate cortex at E14.5, the mice were sacrificed at E17.5, and subjected to immunostaining with L1 and DsRed (Figure 12). The expression of DCC-‐long isoform in cingulate cortex at E14.5 was not sufficient to form the commissure corpus callosum, indicating that the DCC-‐long isoform was not sufficient to rescue the ACC phenotype, and suggesting the possibility that the ACC phenotype in *Nova2-*KO mice either does not involve the DCC-‐long isoform or is caused by the dysregulation of multiple transcriptomes. We mention this negative result in the revised Discussion part.10.7554/eLife.14371.031Author Response Image 1.**DOI:**
http://dx.doi.org/10.7554/eLife.14371.031

Without the DCC or Neo1 rescue experiments, the authors should widen the discussion about the possible targets affected in the Nova2A mutant that could give rise to the agenesis phenotype. There are around 60 mouse genes known to be required for the formation of the corpus callosum (e.g. Paul et al. 2007; Donahoo and Richards, 2012) so it would valuable to try to establish how many of these identified genes are targets of Nova2. How many of these 60 genes are present in the Nova2A target data? Loss of Neuropilin1, for example, gives rise to ACC (Lim et al. 2015) and the EphB1 and EphB2 intracellular signalling domains are required for the formation of the CC (Robichaux et al. 2015).

We appreciate the reviewers’ point, and have widened the discussion about possible targets affected in the *Nova2*-KO that could give rise to the ACC phenotype, based on great review articles (Paul et al., 2007; Edwards, et al., 2014) and the genes that reviewers suggested (*Neuropilin1, EphB1, EphB2, nSR100/Srrm4*). This suggestion prompted us to find 6 additional *Nova2* dependent alternative splicing events on 5 genes that associated with human ACC syndromes and mouse ACC phenotypes (*Ank2, Dclk1, Enah, Cask, Ptprs*;). This has now been added to the revised manuscript (Figure 6—figure supplement 3).

These two studies also report the loss of the anterior commissure in addition to the CC. Does the Nova2 mutant show a defect in axons crossing the AC?

A defect in axons crossing the anterior commissure midline was not observed in *Nova2*-KO mice (5/5 mice) (red arrowin Figure 6—figure supplement 2: DAPI stain at E18.5). This point and data has now been added to the text (Figure 6—figure supplement 2).

2) Overall, the authors do not fully credit highly relevant studies in the literature. In addition to examples above, Blencowe's group published a particularly relevant study with remarkably similar findings to the present work last year that is not cited here (Quesnel-Vallieres et al., G & D 2015). They report that the loss of nSR100, an alternative splicing factor in neurons, gives rise to CC agenesis and to phrenic nerve defects. In addition, a mutation in nSR100 has been reported to cause hearing defects and should probably be cited along with the other two the authors cite in the statement "to date only two alternative splicing factors have been implicated in hearing: Srrm4 (Nakano et al. 2012) and Sfswap (Moayedi et al. 2014)".

We now use *Srrm4/nSR100* to avoid confusion since *Srrm4* and *nSR100* are same gene. The article of *Srrm4/nSR100* exon 7-‐8 deletion mice (Quesnel-‐Vallieres M., et al., (2015) Genes Dev., 29(7):746-‐759) was cited as a relevant study with similar findings in CC and phrenic nerve to our present work; this is now clarified in the revised text. The statement about alternative splicing and hearing was likewise modified to “to date only two alternative splicing factors have been implicated in hearing: *Srrm4/nSR100* (Nakano et al. 2012) and *Sfswap* (Moayedi et al. 2014)”.

3) Given the limitations of the Study, perhaps a more accurate title would be "NOVA-2 mediated alternative splicing is required for axon pathfinding during development." Indeed, the data certainly demonstrate that this is the case. While "regulatory networks" allude to profound insights, to what extend this is true remains unclear, while it is clear that "alternative splicing" is important.

We agree, and have changed the title to “NOVA2-‐mediated RNA regulation is required for axonal pathfinding during development” since although we focused on alternative splicing but NOVA2 also has other actions, including regulation of alternative poly(A) usage (Licataloci et al., (2008) Nature 456:464–69).

4) The authors should provide schematic representations for the developmental systems to make the data more broadly accessible to readers. It would also be helpful to make it clear to the reader that the specific phenotypes are complex and as NOVA-2 is broadly expressed how specific developmental defects arise may be a consequence not only of changes in transcripts in a specific cell type, but also changes in transcripts in different ways in different neurons. That is, the phenotypic analysis assumes cell autonomy and no data addresses this issue in the paper.

We modified the schematic Figure 11 and Figure 11 legend. Also, we widened the discussion about alternative mechanisms (e.g. GO enriched pathway (protein modification, phosphorylation, metabolism and chromatin modification) and differentiation).

Although we could not distinguish the possibility that *Nova2* deficits cause abnormally differentiated neurons as a primary defect and thereby result in axonal pathfinding defects, we do note that our RNAseq data showed that the transcriptome abundance of the neuronal differentiation markers were comparative between wild-‐type and *Nova2*-‐KO mice (see [Supplementary-material SD2-data]). Our previous work (Yano M et al., (2010) Neuron66:848-‐858) has showed that the proliferation of neural progenitor cell (NPC) is not disrupted in *Nova2*-KO, suggesting that the differentiation from NPC to neuron is comparative between wild-‐type and *Nova2*-KO mice at least in the cortex. This point and data has now been added to the text ([Supplementary-material SD2-data]).

5) Furthermore, whether these axon guidance defects arise solely from alteration of transcripts involved in axon guidance or indirectly by disrupting the regulation of other genes involved in neuronal differentiation is unclear. That said, at least some guidance receptors are targets for NOVA-2 regulation and this observation as well as observations in the companion paper provide support for a direct role in this process. It is important to note, however, that the GO analysis presented in the supplementary material [Supplementary-material SD1-data] indicates that other categories of genes are also significantly altered in NOVA-2 mutants, including genes involved protein modification, phosphorlyation, metabolism and chromatin modification. Thus, to what extent the observed phenotypes reflect alterations in axon guidance receptors per se, as opposed to proteins acting to regulate neuronal differentiation more generally and axon guidance only indirectly remains unclear. All this needs to be addressed in the Discussion.

As written in point 4, we widened the Discussion part about these concerns.

6) The significance of the changes in axon guidance receptors is discussed in the context of the long and short forms of DCC, the netrin receptor. This discussion really belongs in the other manuscript by Chen et al., and that focuses specifically on the NOVA-2 regulation of this pathway.

We cross-‐cited our manuscripts with an accompanying paper by Leggere et al. and partially transfer our Discussion part to an accompanying paper by Leggere et al.

Reviewer #1:

The manuscript investigates the functional differences between the two Nova isoforms, Nova1 and Nova2. The authors compare transcriptome-wide HITS-CLIP and RNA-seq data from Nova1 and Nova2 mutants and report evidence showing that Nova2 uniquely regulates the alternative splicing of axon guidance-related genes in cortical development. They also report that loss of Nova2, but not Nova1, gives rise to axonal defects in the cortex, spinal cord and brainstem, particularly in ventral motoneuron axons. Overall, the results indicate that Nova2 controls alternative splicing of some key components involved in axon guidance.

This is an interesting study that compares Nova1 and Nova2 target transcripts and function. The study follows up on previous work from Darnell and colleagues and provides novel evidence for distinct isoform-specific targets and function. The transcriptome-wide analyses are robust and compelling. The functional axonal defects seen with loss of Nova2A are very clear and well described. However, the study does not identify the Nova2A targets that underlie the axon guidance defects. This weakness could potentially be addressed by further experiments.

1) The loss of the corpus callosum is a severe and convincing phenotype in Nova2 mutants. The authors focus their discussion of this phenotype almost entirely on the netrin-1/DCC signaling pathway for the good reason that Nova2 regulates the alternative splicing of two netrin receptors, DCC and Neo1. However, they do not test this directly. The link between the splicing target and the phenotype would be significantly strengthened by the addition of data testing whether DCC or Neo1 can rescue the ACC phenotype. These would be challenging but valuable experiments.

Without the DCC or Neo1 rescue experiments, the authors should widen the discussion about the possible targets affected in the Nova2A mutant that could give rise to the agenesis phenotype. There are around 60 mouse genes known to be required for the formation of the corpus callosum (e.g. Paul et al. 2007; Donahoo and Richards, 2012) so it would valuable to try to establish how many of these identified genes are targets of Nova2. How many of these 60 genes are present in the Nova2A target data? Loss of Neuropilin1, for example, gives rise to ACC (Lim et al. 2015) and the EphB1 and EphB2 intracellular signalling domains are required for the formation of the CC (Robichaux et al. 2015). These two studies also report the loss of the anterior commissure in addition to the CC. Does the Nova2 mutant show a defect in axons crossing the AC?

See answers to point 1 above.

2) Overall, the authors do not fully credit highly relevant studies in the literature. In addition to examples above, Blencowe's group published a particularly relevant study with remarkably similar findings to the present work last year that is not cited here (Quesnel-Vallieres et al., G & D 2015). They report that the loss of nSR100, an alternative splicing factor in neurons, gives rise to CC agenesis and to phrenic nerve defects. In addition, a mutation in nSR100 has been reported to cause hearing defects and should probably be cited along with the other two the authors cite in the statement "to date only two alternative splicing factors have been implicated in hearing: Srrm4 (Nakano et al. 2012) and Sfswap (Moayedi et al. 2014)".

See answer to point 2 above.

Reviewer #2:

This is a provocative paper from Darnell and colleagues. Using a combination of HITS-CLIP and RNA-sequencing, transcripts encoding proteins regulating axon guidance were identified as targets preferentially regulated by the NOVA-2 splicing factor (selectively when compared to NOVA-1). Additional data are presented in support of specific alternative splicing events selectively regulating axon guidance receptors via a NOVA-2 dependent mechanism. In the second part of the paper, genetic studies are presented demonstrating defects in axon guidance in different developmental contexts. The authors argue that these data support the conclusion as stated in their title "NOVA-1 mediated RNA regulatory networks are required for axonal pathfinding during development". Perhaps a more accurate title would be "NOVA-2 mediated alternative splicing is required for axon pathfinding during development." Indeed, the data certainly demonstrate that this is the case. While "regulatory networks" allude to profound insights, to what extend this is true remains unclear, while it is clear that "alternative splicing" is important.

See answer to point 3 above.

The RNA-sequencing and HITS clip data are impressive and comprehensive and reveal an impressive array of transcripts selectively regulated by NOVA-2. Together this makes a compelling argument that NOVA-2 broadly sculpts the neuronal proteome. The genetic data establish marked axon guidance defects in several different developmental contexts and thus broadly supports the notion of a critical role for NOVA-2 in regulating axon guidance. These data are compelling. My only suggestion here is that the authors provide schematic representations for the developmental systems to make the data more broadly accessible to readers. It would also be helpful to make it clear to the reader that the specific phenotypes are complex and as NOVA-2 is broadly expressed how specific developmental defects arise may be a consequence not only of changes in transcripts in a specific cell type, but also changes in transcripts in different ways in different neurons. That is, the phenotypic analysis assumes cell autonomy and no data addresses this issue in the paper.

See answer to point 4 above.

Furthermore, whether these axon guidance defects arise solely from alteration of transcripts involved in axon guidance or indirectly by disrupting the regulating of other genes involved in neuronal differentiation is unclear. That said, at least some guidance receptors are targets for NOVA-2 regulation and this observation as well as observations in the companion paper provide support for a direct role in this process. It is important to note, however, that the GO analysis presented in the supplementary material [Supplementary-material SD1-data] indicates that other categories of genes are also significantly altered in NOVA-2 mutants, including genes involved protein modification, phosphorlyation, metabolism and chromatin modification. Thus, to what extent the observed phenotypes reflect alterations in axon guidance receptors per se, as opposed to proteins acting to regulate neuronal differentiation more generally and axon guidance only indirectly remains unclear.

See answer to point 5 above.

The significance of the changes in axon guidance receptors is discussed in the context of the long and short forms of DCC, the netrin receptor. This discussion really belongs in the other manuscript that focuses specifically on the NOVA-2 regulation of this pathway.

See answer to point 6 above.

In general, I feel that the two papers together certainly are greater than the sum of the two. I suggest that authors play down the notion of a network here as it says very little and implies a level of understanding that is by no means achieved here. Perhaps, the authors could move the discussion of DCC from this manuscript to the other one, and use the space to address the question of how NOVA-2 regulates neural development, and perhaps expand this to explore the relationship to other global splicing regulators of neuronal differentiation such RbFox.

We cross-‐cited our manuscripts with an accompanying paper by Leggere et al. and expanded our Discussion to clarify the contributions of Leggere et al.

Reviewer #3:

This work provides an extensive analysis of loss-of-function phenotypes for NOVA1 and NOVA2 mutant mice. Using a combination of genome-wide approaches the authors identify an alternative splicing program regulated by the NOVA RNA-binding proteins. This analysis extends previously reported data sets on such targets.

Amongst many other targets, there is a significant de-regulation of several axon guidance regulators, in particular components that have been shown to contribute to growth and guidance at the midline. In parallel, the authors report a series of axon extension and guidance phenotypes in the mutant mice.

The datasets are extensive and rigorous. What is a weakness of the work is the lack of data demonstrating a causal link between any of the splicing targets and the anatomical phenotypes.

There is also no insight into how the alternative splice isoforms of the guidance receptors may differ in activity.

The main contributions of the work are the description of novel anatomical phenotypes and splicing targets, as well as the identification of distinct phenotypes in NOVA2 versus NOVA1 mutant mice. However, the mechanism underlying this apparent specificity remains unclear (cell type-specificity of NOVA isoform expression or unique interactions with co-factors or RNA elements?).

Since RNA binding KH domains of both NOVA1 and NOVA2 are identical, the differences in function between NOVA1 and NOVA2 may be derived from the expression, cell milieu, and/or protein-‐protein interaction. As mentioned in the Discussion part largest difference is in spacer domain between KH2 and KH3, although there is no known function of this region. We also widen the Discussion about this point.